# EvoEGF-Mol: Evolving Exponential Geodesic Flow for Structure-based Drug Design

Yaowei Jin [* 1]  Junjie Wang [* 1 2]  Cheng Cao [1]  Penglei Wang [1]  Duo An [1]  Qian Shi [1]

## Abstract

Structure-Based Drug Design (SBDD) aims to discover bioactive ligands. Conventional approaches construct probability paths separately in Euclidean and probabilistic spaces for continuous atomic coordinates and discrete chemical categories, leading to a mismatch with the underlying statistical manifolds. We address this issue by representing molecules using composite exponential-family distributions, where coordinates and categories are represented within a unified natural parameter space to evolve synchronously along exponential geodesics under the Fisher-Rao metric. To avoid the instantaneous trajectory collapse induced by geodesics directly targeting Dirac distributions, we propose Evolving Exponential Geodesic Flow for SBDD (EvoEGF-Mol), which replaces static Dirac targets with dynamically concentrating distributions and is trained with a progressive-parameter-refinement architecture. Our model approaches a reference-level PoseBusters passing rate (93.4%) on CrossDock, demonstrating remarkable geometric precision and interaction fidelity, while achieving superior performance over baseline methods on real-world MolGenBench tasks for bioactive scaffold recovery. Code is available at https://github.com/BLEACH366/EvoEGF-Mol.

## 1. Introduction

Structure-Based Drug Design (SBDD) aims to design bioactive ligands with high binding affinity for specific protein targets(Du et al., 2024; Ferreira et al., 2015). To overcome the vast chemical space and biophysical constraints involved, deep generative models have emerged as an alternative to

---
[*]Equal contribution [1]Lingang Laboratory [2]School of Information Science and Technology, ShanghaiTech University. Correspondence to: Qian Shi <shiqian@lglab.ac.cn>.

*Proceedings of the 43rd International Conference on Machine Learning*, Seoul, South Korea. PMLR 306, 2026. Copyright 2026 by the author(s).

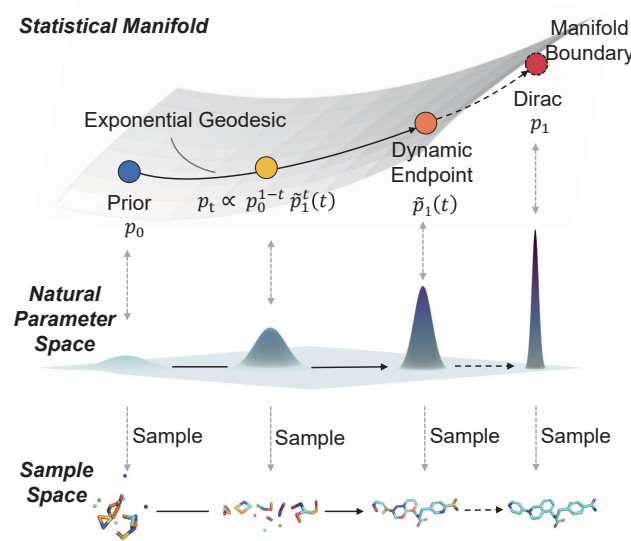

*Figure 1.* The EvoEGF-Mol concept. EvoEGF-Mol generates molecules via e-geodesics on a statistical manifold. To address the singularity at the manifold boundary, we use a Dynamic Endpoint strategy. By defining a time-evolving target (orange circle), the model creates a stable path that gradually concentrates from the prior to the data, guiding the sampling process from noise to valid molecular structures.

brute-force searching, allowing for the direct de novo design of target-specific molecules(Isert et al., 2023; Cremer et al., 2024).

Modeling the joint distribution of continuous atomic positions and discrete topologies remains a central challenge in molecular generation(Qu et al., 2024). Current SBDD methods favor non-autoregressive frameworks, such as Diffusion Models and Flow Matching (FM)(Cremer et al., 2025; Zhang et al., 2024), yet they often use separately designed probability paths for continuous coordinates and discrete atom types. Such designs risk inducing a modality mismatch, where geometric coordinates are driven toward convergence while chemical identities remain weakly determined, thereby failing to capture the structural-chemical synergy essential for effective drug design (Qiu et al., 2025).

We adopt an information-geometric perspective to model molecular data within a unified probabilistic framework. In this formulation, each molecule is represented by a product

of exponential-family components, where different molecular components are assigned appropriate distributional forms. Specifically, atomic coordinates are typically described by Gaussian distributions, while atom and bond types are modeled using categorical or Dirichlet distributions; all of these belong to the exponential family. From this viewpoint, each molecular sample can be viewed as a Dirac distribution, and the full molecular dataset can be interpreted as the collection of these distributions on the statistical manifold. Similarity between molecules is then naturally quantified using the Fisher Information Metric, which provides a principled geometric measure over the underlying distribution space.

A fundamental property of the exponential family is that under the dual information-geometric structure induced by the Fisher–Rao metric and the exponential (e-) connection, exponential geodesics (e-geodesics) correspond to linear interpolation in the natural parameter space within a fixed exponential family. This property provides a geometrically well-motivated probability path for molecular generation. Specifically, continuous atomic coordinates (modeled as Gaussians) and discrete chemical topologies (modeled as Dirichlet distributions) are organized within a product exponential-family manifold using concatenated natural parameters. This formulation allows heterogeneous molecular variables to share a synchronized, linear interpolation schedule defined by the e-geodesics. In this sense, a molecule is represented as a single probabilistic object residing on a Fisher–Rao statistical manifold, rather than as a decoupled combination of Euclidean coordinates and discrete graph structures. Evolving distributions along e-geodesics induces a Fisher-calibrated contraction of uncertainty in atomic identities and spatial configurations, enabling synchronized refinement in a geometrically consistent manner that mitigates the temporal-scale mismatch caused by modalities in conventional methods.

However, directly adopting a Dirac measure as the terminal distribution introduces a practical limitation(Boll et al., 2024). When an e-geodesic is constructed toward a fixed Dirac endpoint, the natural parameters diverge, causing the distribution to collapse rapidly at the early stage of the trajectory and thereby narrowing the effective learning window. Consequently, the model's capacity to capture correlations between chemical identity and spatial geometry is significantly diminished (see Sec. 3.2).

To overcome this limitation, we introduce Evolving Exponential Geodesic Flow for SBDD (EvoEGF-Mol), as shown in Fig. 1. Rather than pursuing a sharply concentrated endpoint from the outset, EvoEGF-Mol employs a dynamic target whose precision increases gradually over time. In this pursuit–evasion process, the resulting generative flow adapts to this evolving target, preventing the

rapid collapse toward a deterministic state and maintaining well-conditioned intermediate distributions while remaining consistent with the Fisher–Rao information geometry. We implement this framework using a progressive-parameter-refinement generative paradigm inspired by Bayesian Flow Networks (BFN)(Graves et al., 2023) and Parameter Interpolation Flow (PIF)(Jin et al., 2025), enabling efficient training and sampling. Furthermore, we demonstrate that the recent Straight-Line Diffusion Model (SLDM)(Ni et al., 2025) is a special case of EGF under specific boundary conditions, and we provide an analysis of their distinctions and connections.

Our contributions can be summarized as follows:

- We introduce a generative framework that synchronously evolves continuous atomic coordinates and discrete chemical categories in a unified natural-parameter space. Adopting e-geodesics as probability paths enables principled alignment with the statistical manifold of chemical and geometric structures.

- We address the numerical instability of e-geodesic probability flows involving Dirac distributions by progressively evolving the target distribution parameters, thereby balancing accuracy and stability in training and generation.

- Experimental results demonstrate that EvoEGF-Mol significantly outperforms traditional SBDD generative models across multiple benchmarks, producing molecules with more realistic geometric structures.

## 2. Related Work

**Structure-Based Drug Design.** SBDD focuses on the de novo generation of 3D molecular structures tailored to a specific protein binding site. Autoregressive methods construct ligands sequentially: early models like AR (Luo et al., 2021) and Pocket2Mol (Peng et al., 2022) focus on atom-by-atom placement, while FLAG (Zhang et al., 2023) constructs ligands by sequentially assembling molecular fragments. PocketFlow (Jiang et al., 2024) incorporate triangular attention into the autoregressive step to better map latent priors to chemical space. To overcome the lack of global coordination for molecule in aforementioned sequential methods, Diffusion-based and Flow Matching models have been introduced and have gained prominence. TargetDiff (Guan et al., 2023a) pioneered the full-atom diffusion framework for SBDD, which was later enhanced by DecompDiff (Guan et al., 2023b) through the integration of explicit chemical priors. D3FG (Lin et al., 2023) lifts this process to the functional group level for enhanced realism. Recent research shifts toward more efficient generative framework: FLOWR (Cremer et al., 2025) integrates continuous and categorical flow matching(Tong et al., 2023; Campbell et al., 2024)

with equivariant optimal transport, DynamicFlow (Zhou et al., 2025) co-models ligand synthesis with protein conformational changes, and ECloudGen (Zhang et al., 2025) leverages electron cloud latent diffusion decoded by LLM architectures to bridge multi-source data gaps. Lastly, unified probabilistic frameworks seek to harmonize discrete atom types with continuous coordinates. MolCRAFT (Qu et al., 2024) introduces BFN(Graves et al., 2023) for joint generation, and MolPilot (Qiu et al., 2025) further optimizes this via VLB-Optimal Scheduling (VOS) to align noise schedules across modalities.

**Flow Matching in Statistical Manifolds.** Recent advancements have extended Flow Matching from Euclidean spaces to complex manifold geometries. For discrete data, SFM (Cheng et al., 2024) and FISHER-FLOW (Davis et al., 2024) reformulate categorical generation by treating probability simplices as statistical manifolds. While SFM leverages the Fisher information metric to derive shortest-path geodesics, FISHER-FLOW utilizes the isometric mapping of the simplex to the positive orthant of a hypersphere to achieve numerically stable, closed-form flows. On structured discrete domains, E-Geodesic FM (Boll et al., 2024) introduces flows on the assignment manifold, utilizing factorizing measures to avoid the pitfalls of heuristic discretization. Beyond modeling individual samples via point-wise manifolds, WFM (Haviv et al., 2025) lifts the FM framework to the space of measures, using Wasserstein geometry to generate families of distributions such as point clouds and Gaussians. Finally, addressing the inference efficiency in curved spaces, the Riemannian Consistency Model (RCM) (Cheng et al., 2025) generalizes consistency models to Riemannian manifolds. By employing covariant derivatives and exponential-map-based parameterizations, RCM enables high-fidelity, few-step generation while strictly maintaining manifold constraints.

## 3. Method

In this section, we introduce EvoEGF-Mol, which formulates molecule generation on the statistical manifold by modeling the transition from a prior distribution to the target Dirac distribution along an evolving e-geodesic path. We first define the SBDD task and introduce e-geodesic in Sec. 3.1. Then, we present EvoEGF in Sec. 3.2. Finally, in Sec. 3.3, we demonstrate how continuous atom coordinates, discrete atom types, and discrete bond types can be modeled within the EvoEGF framework.

### 3.1. Preliminaries

#### 3.1.1. STRUCTURE-BASED DRUG DESIGN (SBDD)

SBDD aims to model the conditional distribution $P(M|P)$ to generate ligand molecules $M$ that bind to a given protein target $P$. The protein binding site is represented as a set of $N_P$ atoms $P = \{(\mathbf{x}_P^{(i)}, \mathbf{v}_P^{(i)})\}_{i=1}^{N_P}$, where $\mathbf{x}_P \in \mathbb{R}^{N_P \times 3}$ and $\mathbf{v}_P \in \mathbb{R}^{N_P \times K_P}$ denote the Cartesian coordinates and discrete atom types, respectively. Similarly, the generated ligand is defined by a triplet $M = (\mathbf{x}_M, \mathbf{v}_M, \mathbf{b}_M)$, where $\mathbf{x}_M \in \mathbb{R}^{N_M \times 3}$ represents the atomic positions, $\mathbf{v}_M \in \mathbb{R}^{N_M \times K_\mathbf{v}}$ encodes the atom types, and $\mathbf{b}_M \in \mathbb{R}^{N_M \times (N_M-1) \times K_\mathbf{b}}$ denotes the discrete bond types, with $K_\mathbf{v}$ and $K_\mathbf{b}$ representing the number of atom and bond categories. The number of ligand atoms $N_M$ is typically sampled from an empirical distribution or predicted by a neural network and is not directly involved in the diffusion process. Consequently, the SBDD task is formulated as learning the joint distribution $P(\mathbf{x}_M, \mathbf{v}_M, \mathbf{b}_M|\mathbf{x}_P, \mathbf{v}_P)$, requiring the model to capture the intricate spatial and chemical dependencies between the ligand components and the protein target.

#### 3.1.2. THE EXPONENTIAL GEODESICS

In the framework of Information Geometry, a statistical manifold is not treated as a flat Euclidean space. Instead, the geometric structure of manifold is characterized by a Riemannian metric and a pair of affine connections, which together define geodesic paths between probability distributions. The e-geodesic represents a straight line in the natural parameter space with respect to the exponential connection (the e-connection).

For distributions belonging to the exponential family, the e-geodesic possesses a fundamental property: it is equivalent to performing a linear interpolation in the space of natural parameters. Given two density functions $p_0(\mathbf{x})$ and $p_1(\mathbf{x})$, the intermediate distribution $p_t(\mathbf{x})$ along the e-geodesic is defined by the log-linear combination:

$$\log p_t(\mathbf{x}) = (1-t)\log p_0(\mathbf{x}) + t\log p_1(\mathbf{x}) - \psi(t) \quad (1)$$

where $\mathbf{x} \in \mathbb{R}^d$ is the $d$-dimensional variable, and $\psi(t)$ is the log-partition function (cumulant generating function) ensuring the normalization constraint $\int_{\mathbb{R}^d} p_t(\mathbf{x})d\mathbf{x} = 1$. In terms of the natural parameters $\boldsymbol{\eta}$, this trajectory corresponds to the linear path

$$\boldsymbol{\eta}_t = (1-t)\boldsymbol{\eta}_0 + t\boldsymbol{\eta}_1. \quad (2)$$

Geometrically, an e-geodesic is linear in the natural coordinates of the distribution. This path ensures that every intermediate distribution $p_t$ remains a valid member of the exponential family, capturing the intrinsic structural transformation between the initial and terminal states.

For isotropic Gaussian distributions $p_0 \sim \mathcal{N}(\boldsymbol{\mu}_0, \sigma_0^2\mathbf{I})$ and $p_1 \sim \mathcal{N}(\boldsymbol{\mu}_1, \sigma_1^2\mathbf{I})$, the intermediate distribution $p_t \sim \mathcal{N}(\boldsymbol{\mu}_t, \sigma_t^2\mathbf{I})$ along the e-geodesic is characterized by the linear interpolation of their natural parameters. Specifically,

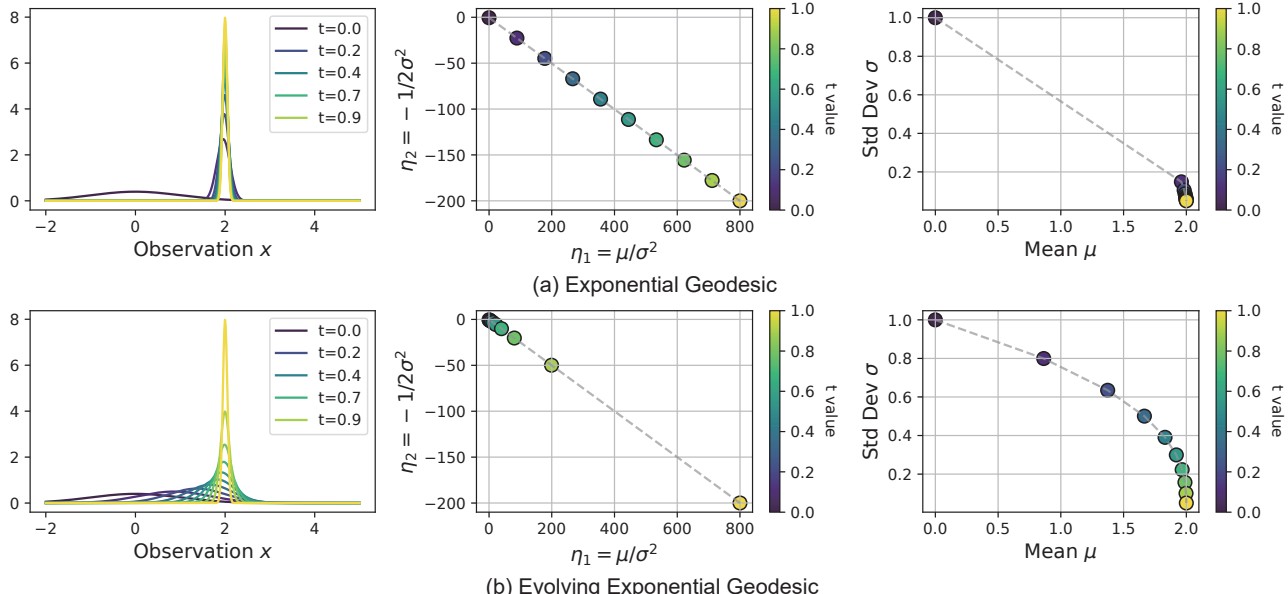

*Figure 2.* Comparison of (a) Exponential Geodesic Flow and (b) Evolving Exponential Geodesic Flow. Both paths transition from a one-dimensional standard Gaussian distribution at $x = 0$ toward a target Dirac distribution at $x = 2$. (Left) Evolution of PDFs over time. (Middle) Trajectories of natural parameters over time. (Right) Trajectories of standard parameters over time. The evolving path prevents the premature variance collapse seen in the standard geodesic, ensuring smoother transition dynamics for molecular generation.

the precision-weighted mean vector $\sigma_t^{-2}\boldsymbol{\mu}_t$ and the negative half of the reciprocal of the variance $-1/2\sigma_t^{-2}$ evolve linearly with respect to $t \in [0,1]$:

$$\boldsymbol{\eta}_{t,1} = \sigma_t^{-2}\boldsymbol{\mu}_t = (1-t)\sigma_0^{-2}\boldsymbol{\mu}_0 + t\sigma_1^{-2}\boldsymbol{\mu}_1 \qquad (3)$$

$$\eta_{t,2} = -1/2\sigma_t^{-2} = (1-t)(-1/2\sigma_0^{-2}) + t(-1/2\sigma_1^{-2}) \quad (4)$$

For Dirichlet distributions, the natural parameters are defined as $\eta_i = \alpha_i - 1$. Substituting these into the e-geodesic framework, the trajectory between $p_0(\boldsymbol{\alpha}_0)$ and $p_1(\boldsymbol{\alpha}_1)$ is characterized by the linear evolution of these parameters:

$$\boldsymbol{\eta}_t = \boldsymbol{\alpha}_t - \mathbf{1} = (1-t)(\boldsymbol{\alpha}_0 - \mathbf{1}) + t(\boldsymbol{\alpha}_1 - \mathbf{1}). \quad (5)$$

Although Gaussian precision and Dirichlet concentration quantify uncertainty in different probabilistic spaces, they both admit a linear evolution in natural-parameter coordinates along the e-geodesic. This shared linear structure provides a common information-geometric schedule for heterogeneous variables, enabling a consistent treatment of uncertainty across different types of exponential-family distributions.

### 3.2. Evolving Exponential Geodesic Flow

A critical challenge in path construction arises when the target $p_1$ is treated as a Dirac delta distribution $\delta_{\mathbf{x}^*}$. As illustrated in Fig. 2, this places the target on the boundary of the statistical manifold, leading to a mathematical singularity. In Gaussian-based continuous paths, this manifests as a

collapse where the variance vanishes almost instantaneously as $t \to 1$. Similarly, in Dirichlet-parameterized discrete simplices, it results in "support vanishing", where the probability mass abruptly concentrates onto a single vertex. These phenomena compress the effective training signal into a disproportionately narrow time interval, depriving the model of the intermediate states necessary to learn complex structural correlations. See Appendix B for the detailed derivation of these edge cases.

To address the path degeneracy caused by static Dirac targets, we propose the Evolving Exponential Geodesic Flow (EvoEGF). Instead of constructing a geodesic towards a fixed, singular boundary point, EvoEGF establishes a probability path towards a dynamically concentrating distribution. This ensures that the generative trajectory remains within the interior of the statistical manifold, thereby maintaining a well-defined probability path and guaranteeing an effective learning window for model training.

**Exponential Family Manifold.** Consider a minimal exponential family $\mathcal{P} = \{p(\cdot|\boldsymbol{\eta}) : \boldsymbol{\eta} \in \Omega\}$ defined on domain $\mathcal{X}$, where $\Omega \subseteq \mathbb{R}^d$ is the open convex space of natural parameters. The density is given by:

$$p(\mathbf{x} \mid \boldsymbol{\eta}) = h(\mathbf{x})\exp\left(\langle \boldsymbol{\eta}, \mathbf{T}(\mathbf{x})\rangle - A(\boldsymbol{\eta})\right), \qquad (6)$$

where $\mathbf{T}(\mathbf{x})$ denotes sufficient statistics and $A(\boldsymbol{\eta})$ is the log-partition function. The manifold $\mathcal{P}$ is equipped with a Riemannian metric defined by the Fisher Information Matrix (FIM), $\mathbf{G}(\boldsymbol{\eta}) = \nabla^2 A(\boldsymbol{\eta})$. Geodesics under the

e-connection (exponential connection) correspond to linear interpolations in $\boldsymbol{\eta}$.

**The Dynamic Target Trajectory.** Let the prior be $\boldsymbol{\eta}_0$ and the clean data be represented by a boundary parameter $\boldsymbol{\eta}_1$ (e.g., infinite precision). We construct a time-evolving target $\tilde{\boldsymbol{\eta}}_1(t)$ through a smooth parameter $\lambda$; the specific smooth form can be found in Sec. 3.3. The generative path is defined as the e-geodesic interpolation:

$$\boldsymbol{\eta}_t = (1-t)\boldsymbol{\eta}_0 + t\tilde{\boldsymbol{\eta}}_1(t). \tag{7}$$

This dynamic-endpoint formulation goes beyond nonlinear time reparameterization of fixed-endpoint paths, defining a richer class of probability paths; see Appendix D. This formulation ensures the path remains in the Riemannian interior $\Omega$ for $t < 1$. Differentiating Eq. 7 yields the tangent vector (parameter velocity) $\dot{\boldsymbol{\eta}}_t$:

$$\dot{\boldsymbol{\eta}}_t = (\tilde{\boldsymbol{\eta}}_1(t) - \boldsymbol{\eta}_0) + t\dot{\tilde{\boldsymbol{\eta}}}_1(t). \tag{8}$$

This parameter evolution defines a tangent vector on the exponential-family statistical manifold. For $p_t(\mathbf{x}) = p(\mathbf{x} \mid \boldsymbol{\eta}_t)$, differentiating the log density gives

$$\partial_t p_t(\mathbf{x}) = p_t(\mathbf{x})\big(\mathbf{T}(\mathbf{x}) - \nabla A(\boldsymbol{\eta}_t)\big)^{\top}\dot{\boldsymbol{\eta}}_t. \tag{9}$$

The Fisher-Rao norm of this statistical tangent is therefore

$$\|\partial_t p_t\|_{\mathrm{FR},p_t}^2 = \dot{\boldsymbol{\eta}}_t^{\top}\mathbf{G}(\boldsymbol{\eta}_t)\dot{\boldsymbol{\eta}}_t, \quad \mathbf{G}(\boldsymbol{\eta}_t) = \nabla^2 A(\boldsymbol{\eta}_t). \tag{10}$$

This parameter-space evolution directly defines our dynamic path on the statistical manifold. While such a path implicitly induces a sample-space continuous normalizing flow via the continuity equation, directly parameterizing and optimizing non-unique sample-space velocity fields for joint discrete-continuous molecular distributions presents notable challenges. Therefore, in Sec. 3.3, we propose a streamlined training framework that directly operates on the parameter space, inspired by BFN and PIF. The same Fisher-Rao metric that measures the instantaneous tangent in Eq. 10 also measures the local refinement error used by our training objective. The optional sample-space characterization and its local Fisher-Rao interpretation are given in Appendices H.2 and H.3.

We demonstrate that SLDM is a special case of our EGF framework under regularized static endpoints, whereas our EvoEGF provides a generalized dynamic solution to the intrinsic endpoint singularity (see Appendix E).

### 3.3. Molecular Generation via EvoEGF

We apply EvoEGF to SBDD, generating a ligand $M$ conditioned on a protein pocket $P$. The prior and intermediate distributions are modeled as products of exponential families.

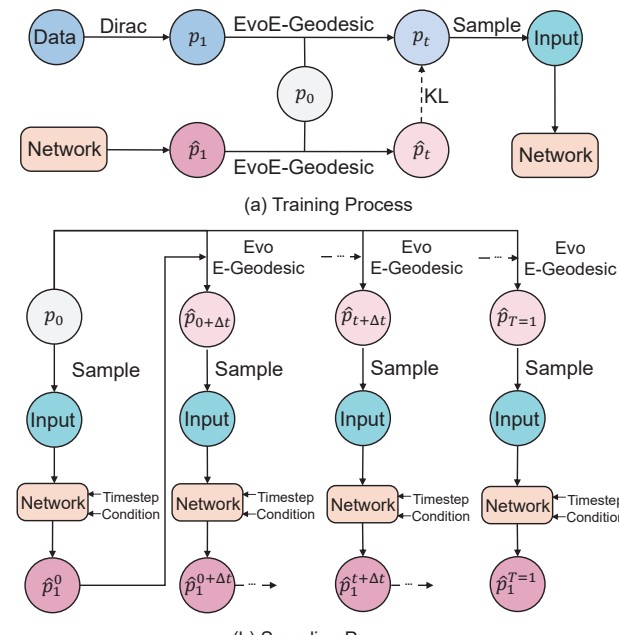

(a) Training Process

(b) Sampling Process

*Figure 3.* The training and sampling process of EvoEGF-Mol. EvoEGF-Mol trains a neural network to map noisy intermediate states to terminal parameters by minimizing KL divergence along evolving e-geodesic. During inference, molecular structures are generated through iterative refinement, with look-ahead predictions guiding state transport from prior to data distribution. See Appendix A for details.

**Generative Procedure.** EvoEGF-Mol operates via progressive parameter refinement. The training and sampling process are shown in Fig. 3. **(1) Training:** Given a data pair $(M, P)$, we sample a time $t \sim \mathcal{U}(0,1)$ and compute the intermediate parameters $\boldsymbol{\eta}_t$ using the analytical schedule (Eq. 7). We sample noisy states $M_t \sim p(\cdot|\boldsymbol{\eta}_t)$ and train a neural estimator $\boldsymbol{\Phi}(M_t, t, P)$ to predict the terminal parameters $\hat{\boldsymbol{\eta}}_1$. The loss minimizes the local transport error between the true evolution and the model prediction. **(2) Sampling:** Starting from prior samples $M_0 \sim p(\cdot|\boldsymbol{\eta}_0)$, At each step, we predict $\hat{\boldsymbol{\eta}}_1$, and then $\hat{\boldsymbol{\eta}}_t$ is obtained by combining $\hat{\boldsymbol{\eta}}_1$ and $\boldsymbol{\eta}_0$. The sample $\hat{M}_t \sim p(\cdot|\hat{\boldsymbol{\eta}}_t)$ is used as input of the neural network. After iterating the above procedure for a certain number of steps, the result of molecule is sampled at $t = 1$.

**Fisher-Calibrated Synchronized Refinement.** EvoEGF-Mol refines coordinates, atom types, and bond types on a product exponential-family manifold, rather than by separately tuned Euclidean and categorical objectives. The network predicts terminal natural parameters and is trained by the one-step discrepancy between the true evolving path and the model-induced next distribution. Let $\boldsymbol{\eta}_t = (\boldsymbol{\eta}_t^{\mathbf{x}}, \boldsymbol{\eta}_t^{\mathbf{v}}, \boldsymbol{\eta}_t^{\mathbf{b}})$ denote the concatenated natural parameters for coordinates, atom types, and bond types, respectively. Under the product assumption, the full Fisher information matrix is block diagonal. Therefore, for a small local transport

error $\boldsymbol{\xi}_t = \widehat{\boldsymbol{\eta}}_{t+\Delta t} - \boldsymbol{\eta}_{t+\Delta t}$, the KL objective admits the Fisher-Rao quadratic expansion

$$D_{\mathrm{KL}}\big(p_{\boldsymbol{\eta}_{t+\Delta t}} \,\|\, p_{\widehat{\boldsymbol{\eta}}_{t+\Delta t}}\big) = \frac{1}{2}\sum_{\mathbf{c}\in\{\mathbf{x},\mathbf{v},\mathbf{b}\}}(\boldsymbol{\xi}_t^{\mathbf{c}})^{\top}\mathbf{G}^{\mathbf{c}}(\boldsymbol{\eta}_t^{\mathbf{c}})\boldsymbol{\xi}_t^{\mathbf{c}} \\ + o(\|\boldsymbol{\xi}_t\|^2), \tag{11}$$

where $\mathbf{G}^{\mathbf{c}}(\boldsymbol{\eta}_t^{\mathbf{c}}) = \nabla^2 A^{\mathbf{c}}(\boldsymbol{\eta}_t^{\mathbf{c}})$ is the Fisher information matrix of the corresponding exponential-family component. Therefore, the relative strengths of coordinate, atom-type, and bond-type supervision are automatically calibrated by the intrinsic uncertainty of each component, rather than relying on heuristically designed weights between losses across different modalities.

**Coordinate Flow (Isotropic Gaussian).** We define the target schedule via a variance $\tilde{\sigma}_1^2(t)$ decaying according to $\tilde{\sigma}_1(t) = \lambda(1 - t)$. Minimizing the KL divergence along this path is equivalent to learning a denoiser $\hat{\boldsymbol{\mu}}_1(\mathbf{x}_t, t)$. The resulting loss function is a weighted MSE:

$$\mathcal{L}_{\mathbf{x}} = \mathbb{E}_{t,\mathbf{x}^*}\left[\frac{t^2\sigma_t^2}{2\tilde{\sigma}_1^4(t)}\|\mathbf{x}^* - \hat{\mathbf{x}}\|^2\right], \tag{12}$$

where $\sigma_t$ is calculated by Eq. 4.

**Atom and Bond Type Flow (Dirichlet).** Discrete types are mapped to the simplex $\Delta^{K-1}$ using a Dirichlet distribution $\mathrm{Dir}(\boldsymbol{\alpha})$. For atom types $\mathbf{v}$ (and analogously for bond types $\mathbf{b}$), we define the target schedule via a concentration parameter $\tilde{\boldsymbol{\alpha}}_1(t)$ concentrating towards the one-hot label $\mathbf{e}_k$ according to $\tilde{\boldsymbol{\alpha}}_1(t) = (1-\lambda(1-t))\mathbf{e}_k + \lambda(1-t)\frac{1}{K}\mathbf{1}_K$. The training objective minimizes the KL divergence between the one-step-ahead distribution $p_t$ and the model approximation $\hat{p}_t$. For the Dirichlet manifold, this yields:

$$\mathcal{L}_{\mathrm{type}} = \mathbb{E}\left[\ln\frac{\mathcal{B}(\boldsymbol{\alpha}_t)}{\mathcal{B}(\hat{\boldsymbol{\alpha}}_t)} + \sum_k(\hat{\alpha}_{t,k} - \alpha_{t,k})\Delta\psi_k(\hat{\boldsymbol{\alpha}}_t)\right], \tag{13}$$

where $\mathcal{B}(\cdot)$ is the multivariate Beta function, $\psi$ is the digamma function, and $\Delta\psi_k(\boldsymbol{\alpha}) = \psi(\alpha_k) - \psi(\sum_j \alpha_j)$.

# 4. Experiments

## 4.1. Experimental Setup

We conduct two primary experiments on de novo molecular design in SBDD: (1) an empirical evaluation on CrossDock using computational metrics to assess generative quality, and (2) a real-world applicability study on MolGenBench focusing on practical drug discovery–oriented metrics.

**Dataset.** Consistent with previous benchmarks (Luo et al., 2021; Qu et al., 2024), we utilize CrossDock (Francoeur et al., 2020), filtered for high-quality poses (RMSD < 1Å) and 30% sequence identity. This yields 100,000 training

pairs and a test set of 100 unseen proteins (100 samples per target). To assess real-world utility, we also employ MolGenBench (Cao et al., 2025). Its De Novo Design suite includes 120 diverse targets and 220,005 validated active molecules (IC50/Ki/Kd $\leq 10\mu$M), mitigating protein family bias. We sample 1,000 candidates per target for this evaluation.

**Baselines.** We compare our approach with five SBDD baselines across three categories: (1) Autoregressive-based: AR (Luo et al., 2021) and Pocket2Mol (Peng et al., 2022); (2) Diffusion-based: TargetDiff (Guan et al., 2023a) and DecompDiff (Guan et al., 2023b); and (3) BFN-based: MolCRAFT (Qu et al., 2024).

**Metrics.** For the CrossDock dataset, we evaluate our method from multiple perspectives, including pose validity, binding affinity, conformation stability, and molecular properties. Pose validity is measured by the PoseBusters passing rate (PBValid)(Buttenschoen et al., 2024). Binding affinity is assessed using AutoDock Vina(Eberhardt et al., 2021), reportingVina Score (directly scoring the generated pose), Vina Min (after local energy minimization), and Vina Dock (re-docking to estimate the optimal achievable affinity). Conformation stability is evaluated using Strain Energy, which reflects the energetic cost of the ligand conformation relative to its intrinsic low-energy state (calculated by PoseCheck(Harris et al., 2023)). Generation success is quantified by the percentage of fully connected molecules (Connected). Molecular properties are assessed using QED(Quantitative Estimate of Drug-likeness) and SA(Synthetic Accessibility). For fairness, all samples were evaluated using the same metric computation pipeline.

For the MolGenBench, we focus on scaffold-level evaluation, reflecting early-stage drug discovery. Chemical validity is measured by the chemical filter(Schuffenhauer et al., 2020; Bruns & Watson, 2012; Gaulton et al., 2012) pass rate, ensuring generated scaffolds satisfy standard medicinal chemistry rulesets. The model's ability to recover bioactive chemical space is assessed using scaffold-based Hit Rediscover metrics: Hit Recovery, which measures whether at least one known active scaffold is recovered for a target, and Hit Fraction, which quantifies the coverage of known active scaffolds. Target specificity is measured by the Target-Aware Score (TAScore), comparing scaffold recovery under target-aware generation versus a background baseline. Results are reported for both seen and unseen targets to assess generalization.

## 4.2. De novo Design

**EvoEGF-Mol achieves accurate molecular geometries.** The metrics is shown in Table 1. For intramolecular validity, the quartiles of the strain energy of molecules generated by EvoEGF-Mol are all within single-digit or double-digit

*Table 1.* Performance on CrossDock, where EvoEGF-Mol shows robust results. (↑) / (↓) indicates larger / smaller is better. The top-2 results are **bolded** and underlined. Note: CR is short for Clash Ratio. Baselines are evaluated based on released samples from Qiu et al. (2025).

| Methods | PB-Valid Avg. (↑) | Vina Score (↓) Avg. | Vina Score (↓) Med. | Vina Min (↓) Avg. | Vina Min (↓) Med. | Vina Dock (↓) Avg. | Vina Dock (↓) Med. | Strain Energy (↓) 25% | Strain Energy (↓) 50% | Strain Energy (↓) 75% | Connected Avg. (↑) | QED Avg. (↑) | SA Avg. (↑) | CR Avg. (↓) |
|---|---|---|---|---|---|---|---|---|---|---|---|---|---|---|
| CrossDock | 95.0% | -6.36 | -6.46 | -6.71 | -6.49 | -7.45 | -7.26 | 34 | 107 | 196 | - | 0.48 | 0.73 | 0.17 |
| AR | 59.0% | -5.75 | -5.64 | -6.18 | -5.88 | -6.75 | -6.62 | 259 | 595 | 2286 | 93.5% | 0.51 | 0.63 | **0.22** |
| Pocket2Mol | 72.3% | -5.14 | -4.70 | -6.42 | -5.82 | -7.15 | -6.79 | 102 | 189 | 374 | 96.3% | **0.57** | **0.76** | 0.56 |
| TargetDiff | 50.5% | -5.47 | -6.30 | -6.64 | -6.83 | **-7.80** | **-7.91** | 369 | 1243 | 13871 | 90.4% | 0.48 | 0.58 | 0.53 |
| DecompDiff | 71.7% | -5.19 | -5.27 | -6.03 | -6.00 | -7.03 | -7.16 | 115 | 421 | 1424 | 82.9% | 0.51 | 0.66 | 0.51 |
| MolCRAFT | 84.6% | **-6.55** | **-6.95** | **-7.21** | **-7.14** | -7.67 | -7.82 | 83 | 195 | 510 | 96.7% | 0.50 | 0.67 | 0.26 |
| EvoEGF-Mol | **93.4%** | -6.14 | -6.89 | -6.98 | -7.12 | -7.72 | -7.88 | **8.94** | **25.96** | **56.65** | **98.6%** | 0.53 | 0.75 | 0.24 |

*Table 2.* Performance on MolGenBench regarding scaffolds metrics. Definitions: **In** (Proteins in CrossDock), **In(RM.)** (Proteins in CrossDock (remove scaffolds in CrossDock train set)), **Not** (Proteins not in CrossDock). (↑) / (↓) indicates larger / smaller is better. The top-2 results are **bolded** and underlined. Baselines are evaluated based on released samples from Cao et al. (2025).

| Methods | Pass Rate (↑) In (%) | Pass Rate (↑) Not (%) | Hit Recovery (↑) In | Hit Recovery (↑) In(RM.) | Hit Recovery (↑) Not | Hit Rate (↑) In (%) | Hit Rate (↑) In(RM.) (%) | Hit Rate (↑) Not (%) | Hit Fraction (↑) In (%) | Hit Fraction (↑) In(RM.) (%) | Hit Fraction (↑) Not (%) | TAScore Count In 0-1 | TAScore Count In 1-10 | TAScore Count In 10-100 | TAScore Count In >100 | TAScore Count Not 0-1 | TAScore Count Not 1-10 | TAScore Count Not 10-100 | TAScore Count Not >100 |
|---|---|---|---|---|---|---|---|---|---|---|---|---|---|---|---|---|---|---|---|
| Pocket2Mol | 11.58 | 11.10 | 57 | 55 | 29 | 1.75 | 1.17 | 2.03 | 0.61 | 0.48 | 0.53 | 54 | 31 | 0 | 0 | 24 | 11 | 0 | 0 |
| TargetDiff | 5.82 | 6.48 | 62 | 56 | 32 | 1.73 | 1.09 | 2.07 | 0.45 | 0.34 | 0.43 | 38 | 45 | 2 | 0 | 20 | 15 | 0 | 0 |
| DecompDiff | 16.11 | 16.52 | 65 | 61 | 28 | **4.99** | **2.63** | **3.86** | 0.81 | 0.68 | 0.64 | 46 | 35 | 1 | 0 | 20 | 10 | 0 | 0 |
| MolCRAFT | 23.07 | 23.65 | 66 | 64 | **33** | 3.50 | 1.89 | 2.87 | 0.70 | 0.57 | 0.73 | 52 | 33 | 0 | 0 | 22 | 13 | 0 | 0 |
| EvoEGF-Mol | **31.79** | **29.61** | **72** | **72** | **33** | 3.88 | 2.20 | 3.51 | **1.08** | **0.88** | **0.93** | 52 | 30 | 3 | 0 | 24 | 11 | 0 | 0 |

values, significantly outperforming the baseline, as shown in Fig. 4. In addition, Fig. 9, 10, 11, further demonstrate that EvoEGF-Mol can accurately capture the common patterns of bond length, bond angle, and torsion angle distributions observed in the CrossDock dataset. Intermediate-state analysis shows that molecules generated by EvoEGF-Mol form chemically meaningful prototypical structures at an earlier stage, with better synchronized refinement between coordinates and topology; for further details, refer to Appendix J.4.

**EvoEGF-Mol generates improved binding poses.** In terms of binding affinity, EvoEGF-Mol achieves competitive performance across all Vina-based metrics. Moreover, its PB-Valid score substantially surpasses those of other baselines, ensuring reasonable binding poses and demonstrating strong protein–ligand interaction validity.

**EvoEGF-Mol improves real-world scaffold discovery but highlights remaining limitations.** The metrics are shown in Table 2. We evaluate EvoEGF-Mol on MolGenBench with a scaffold-level focus to assess its ability to discover diverse chemical series. As shown in Table 2, EvoEGF-Mol consistently outperforms baselines across all scaffold metrics. It achieves the highest Scaffold Pass Rate (31.79% for CrossDock targets and 29.61% for non-CrossDock targets), indicating strong compliance with medicinal chemistry filters. Moreover, EvoEGF-Mol recovers the largest number of bioactive scaffolds (72 and 33 targets, respectively) and attains the highest Scaffold Hit Fraction (1.08% and 0.93%). Nonetheless, absolute performance remains low across methods (e.g., Scaffold Hit Rates < 5%), indicating that improved geometric plausibility provides a nec-

essary foundation for hit discovery, but alone cannot bridge the efficiency gap required for practical drug discovery.

### 4.3. Ablation Studies

To validate the design choices in EvoEGF-Mol, we conduct an ablation study on the evolving strategy, Dirichlet parameterization, smoothing schedule, explicit bond generation and sampling steps (Fig. 5). We evaluate PoseBuster Validity (PB-Valid), binding affinity (Vina Score / Min / Dock), and Strain Energy (SE). Full results are reported in Appendix J.2.

**Evolving E-Geodesic mechanism.** Replacing the proposed evolving endpoint strategy with a naïve e-geodesic using a fixed Dirac target leads to severe degradation: low PB-Valid, high SE, and repulsive Vina scores. This indicates that prematurely collapsing the target variance or concentration induces excessively imbalanced training signals and prevents effective convergence. In contrast, dynamically evolving the endpoint variance and concentration yields substantial improvements across all metrics, demonstrating that the evolving strategy is essential for stable exponential-geodesic generation on the Fisher–Rao manifold.

**Dirichlet parameterization.** We observe that aggressive early concentration toward the target atom type, as in standard Dirichlet Flow Matching, restricts exploration and results in higher strain energy and reduced validity. A uniform-to-one-hot schedule improves performance but remains inferior to our proposed parameterization. These results highlight the importance of a carefully designed Dirichlet evolu-

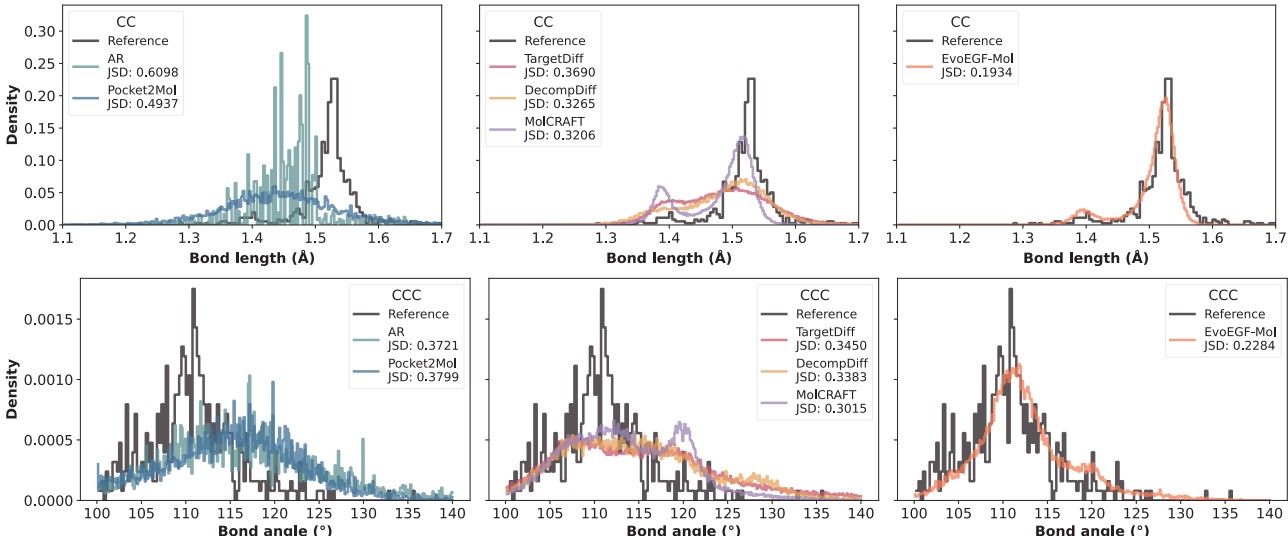

*Figure 4.* Bond length and bond angle distributions for the most common bond types in the CrossDock test set. Results for additional bond types are provided in Appendix J. Our EvoEGF-Mol consistently exhibits improved accuracy in reproducing molecular geometries.

tion to balance exploration and categorical determinism.

**Smoothing coefficients.** Varying the smoothing coefficient shows an optimum around $\lambda = 0.2$, balancing training stability and geometric precision. Smaller $\lambda$ values approach a static Dirac endpoint, causing trajectory collapse, while larger values over-smooth the endpoint and degrade geometric quality.

**Bond diffusion.** Training the model without explicit bond diffusion and reconstructing bonds only post-hoc leads to lower validity and higher strain energy, despite competitive Vina scores. This suggests that coordinate diffusion alone can identify favorable binding regions, whereas explicit bond modeling during training is crucial for enforcing chemical constraints and preserving molecular integrity. Even in this bond-free setting, EvoEGF-Mol consistently outperforms TargetDiff and MolCRAFT under the same training data and architecture, highlighting the inherent advantages of the proposed framework.

**Sampling steps.** We further evaluated our model's robustness under reduced sampling steps ($n$). When $n = 50$, the model still achieves $92.0\%$ PB-Valid while maintaining low strain energy; notably, it outperforms baselines even with only $n = 20$.

### 4.4. Case Study

Taking BSD as a representative case, reference and generated molecules are shown in Fig. 6, The reference molecule forms a stable binding configuration with the protein through a network of interactions with ASP26, ALA55, GLU56, ARG82, and SER86. Among these, the interactions with ALA55 and GLU56 are particularly critical. In

the generated molecule (ID: 12), the interactions with these two key residues are maintained, and the hydrogen bond with ARG82 is successfully preserved. Compound 12 establishes novel interactions with residues located in the center of the pocket. Specifically, the chlorine-substituted six-membered ring in Compound 12 undergoes $\pi$-$\pi$ stacking with PHE49, and the chlorine atom forms a robust interaction—potentially a halogen bond or a reinforced contact—with THR62. Such interactions may enhance the binding specificity of Compound 12. A similar trend is observed in the CD38 case study. Compared to the reference ligands, the generated molecule tends to retain key interactions while creatively engaging with residues that the reference molecules fail to occupy. Nevertheless, we noted that certain drug-likeness features in some generated molecules warrant further optimization.

## 5. Conclusions

We propose EvoEGF-Mol, a generative framework for SBDD that reformulates molecular generation through the lens of information geometry. By defining flows along evolving e-geodesic under the Fisher-Rao metric, EvoEGF-Mol enables a synchronized generative process across multimodal spaces while preventing trajectory collapse. Extensive experiments demonstrate its superior molecular geometric plausibility and binding quality. Building on this geometric foundation, future work will focus on scaling EvoEGF-Mol with larger and more diverse bioactive datasets, with the goal of further validating its utility in real-world drug design pipelines.

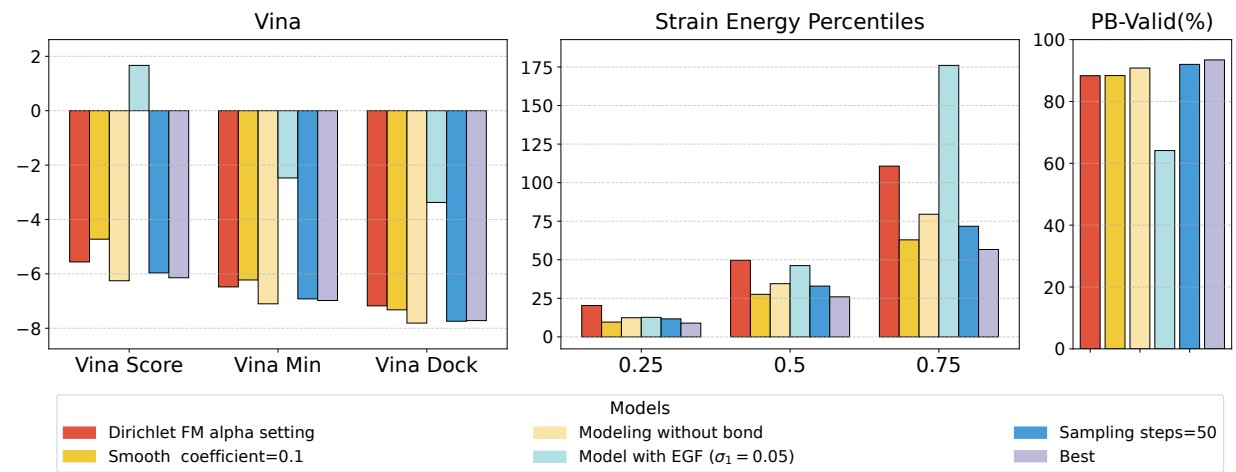

*Figure 5.* Ablation study on Dirichlet parameter settings, the Dirichlet smooth vector, smooth coefficients, bond generation, and the effectiveness of EvoEGF, in terms of binding affinity, strain energy, and PB-Valid.

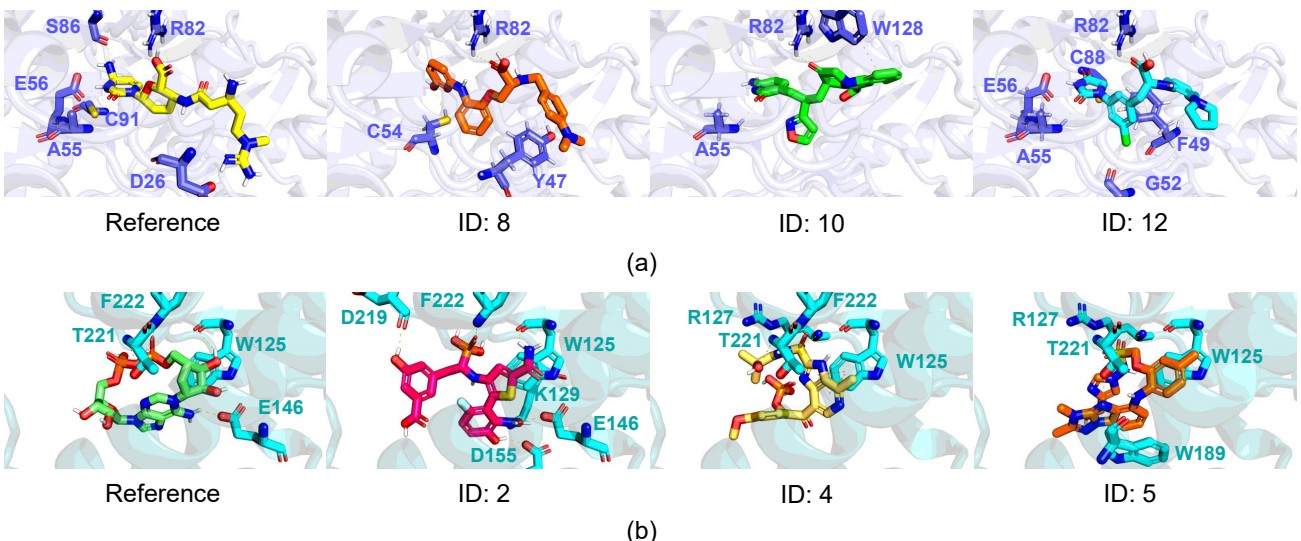

*Figure 6.* Binding mode analysis of generated molecules in BSD and CD38.

## Impact Statement

This paper is aimed to facilitate in-silico structure-based drug design through information-geometric generative flows. Potential societal consequences include the risk of mal-intended usage for discovering toxic or harmful compounds. However, such outcomes require substantial validation in professional wet labs, making them difficult and expensive to realize. Therefore, we believe the primary impact of this work is positive—offering more stable and geometrically precise molecular generation—while remaining vigilant regarding its potential ethical implications.

## Acknowledgments

This work is supported by the Shanghai Rising-Star Program (23QD1400600) and Eastern Talent Plan (QNZH2025122).

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

# A. Methods Details

EvoEGF-Mol follows the progressive-parameter-refinement generative framework introduced in BFN (Graves et al., 2023) and PIF (Jin et al., 2025), but uses an evolving e-geodesic in natural-parameter coordinates. This results in a different probability path while remaining compatible with the same generative structure.

## A.1. Training and Inference

During training, the model learns a neural parameter estimator $\mathbf{\Phi}$ that maps noisy intermediate molecular states to terminal distribution parameters. For a protein-ligand pair $(P, M)$, we sample a time $t \sim \mathcal{U}[0, 1]$ and construct time-dependent target parameters $\hat{\boldsymbol{\theta}}_1(t)$ according to the predefined schedules of $\tilde{\sigma}_1(t)$ and $\tilde{\boldsymbol{\alpha}}_1(t)$. Together with the prior parameters $\boldsymbol{\theta}_0$, these define an intermediate distribution $p_t$ along the e-geodesic. A noisy sample $M_t \sim p_t$ is then fed into $\mathbf{\Phi}(M_t, t, P)$ to predict the terminal parameters $\hat{\boldsymbol{\theta}}_1$. The training objective minimizes the KL divergence between the ground-truth distribution $p_t$ and the model-induced distribution $\hat{p}_t$, enforcing local consistency of probability transport along the geodesic.

At inference time, generation proceeds by iteratively transporting samples from the prior distribution $p_0$ to the terminal distribution. Starting from $M_0 \sim p_0$, the model predicts $\hat{\boldsymbol{\theta}}_1$ at each time step and updates the distribution parameters by moving along the e-geodesic toward the predicted target while following the evolving schedules. This process is repeated until $t = 1$, at which point a final sample is drawn as the generated molecule. This procedure defines a fully generative process that does not require score estimation or explicit likelihood gradients.

---

**Algorithm 1** EvoEGF-Mol Training

---

**Require:** Ligand $M = (\mathbf{x}_M, \mathbf{v}_M, \mathbf{b}_M)$, Protein $P = (\mathbf{x}_P, \mathbf{v}_P)$, Steps $n \in \mathbb{N}^+$, Network $\mathbf{\Phi}$, Learning rate $r$, Atom type number $K_{\mathbf{v}}$, Bond type number $K_{\mathbf{b}}$
1: $i \sim U(1, n), t \leftarrow \frac{i-1}{n}, \epsilon \leftarrow 10^{-6}, \lambda \leftarrow 0.2, \boldsymbol{\alpha}_0 \leftarrow \mathbf{1}$
2: // *Evolving Gaussian Geodesic for Coordinates*
3: $\tilde{\sigma}_1(t) \leftarrow \lambda(1 - t) + \epsilon$
4: $\sigma_t \leftarrow \left((1 - t)/\sigma_0^2 + t/\tilde{\sigma}_1(t)^2\right)^{-1/2}$
5: $\boldsymbol{\mu}_t \leftarrow \sigma_t^2 \left(\frac{(1-t)\boldsymbol{\mu}_0}{\sigma_0^2} + \frac{t\mathbf{x}_M}{\tilde{\sigma}_1(t)^2}\right)$ {Note: $\boldsymbol{\mu}_0 = \mathbf{0}, \sigma_0 = 1$}
6: $M_{\mathbf{x}} \sim \mathcal{N}(\boldsymbol{\mu}_t, \sigma_t^2 \mathbf{I})$
7: // *Evolving Dirichlet Geodesic for Types*
8: $\tilde{\boldsymbol{\alpha}}_1^{\mathbf{v}}(t) \leftarrow (1 - \lambda(1 - t))\mathbf{v}_M + \lambda(1 - t)\frac{1}{K_{\mathbf{v}}}\mathbf{1}_{K_{\mathbf{v}}}$
9: $\boldsymbol{\alpha}_t^{\mathbf{v}} \leftarrow (1 - t)\boldsymbol{\alpha}_0 + t\tilde{\boldsymbol{\alpha}}_1^{\mathbf{v}}(t)$
10: $\tilde{\boldsymbol{\alpha}}_1^{\mathbf{b}}(t) \leftarrow (1 - \lambda(1 - t))\mathbf{b}_M + \lambda(1 - t)\frac{1}{K_{\mathbf{b}}}\mathbf{1}_{K_{\mathbf{b}}}$
11: $\boldsymbol{\alpha}_t^{\mathbf{b}} \leftarrow (1 - t)\boldsymbol{\alpha}_0 + t\tilde{\boldsymbol{\alpha}}_1^{\mathbf{b}}(t)$
12: $M_{\mathbf{v}} \sim \mathrm{Dir}(\boldsymbol{\alpha}_t^{\mathbf{v}}), M_{\mathbf{b}} \sim \mathrm{Dir}(\boldsymbol{\alpha}_t^{\mathbf{b}})$
13: // *Network Prediction and Loss*
14: $\hat{\mathbf{x}}_M, \hat{\mathbf{v}}_M, \hat{\mathbf{b}}_M \leftarrow \mathbf{\Phi}(M_{\mathbf{x}}, M_{\mathbf{v}}, M_{\mathbf{b}}, P, t)$
15: $\mathcal{L}_{\mathbf{x},t} \leftarrow \frac{t^2 \sigma_t^2 \|\mathbf{x}_M - \hat{\mathbf{x}}_M\|^2}{2\tilde{\sigma}_1(t)^4}$
16: $\mathcal{L}_{\mathbf{v},t} \leftarrow \ln \frac{B(\boldsymbol{\alpha}_t^{\mathbf{v}})}{B(\hat{\boldsymbol{\alpha}}_t^{\mathbf{v}})} + \sum_{j=1}^{K_{\mathbf{v}}} (\hat{\alpha}_{t,j}^{\mathbf{v}} - \alpha_{t,j}^{\mathbf{v}})(\psi(\hat{\alpha}_{t,j}^{\mathbf{v}}) - \psi(\sum \hat{\alpha}_{t,j}^{\mathbf{v}}))$
17: $\mathcal{L}_{\mathbf{b},t} \leftarrow \mathrm{DirichletLoss}(\boldsymbol{\alpha}_t^{\mathbf{b}}, \hat{\boldsymbol{\alpha}}_t^{\mathbf{b}})$ {Same form as $\mathcal{L}_{\mathbf{v},t}$}
18: $\mathbf{\Phi} \leftarrow \mathbf{\Phi} - r\nabla_{\mathbf{\Phi}}(\mathcal{L}_{\mathbf{x},t} + \mathcal{L}_{\mathbf{v},t} + \mathcal{L}_{\mathbf{b},t})$

---

## A.2. Implementation Details

We use unitransformer with edge (Qiu et al., 2025) as the base model, and add gated attention for node, position and bond update. We set the kNN parameter to 32, counts of model layers to 4 and hidden dimension to 128. We use Adam optimizer with learning rate 5e-4, batch size of 16. The model reaches convergence in approximately 16 hours on one A100 GPU. With 100 sampling steps, it achieves inference costs comparable to MolCRAFT, generating 1.5 molecules per second on an RTX 4090. We construct atom-level and edge-level feature representations for protein–ligand complexes. At the atom level, each protein atom is encoded by a one-hot representation of its elemental type (H, C, N, O, S, Se), a 20-dimensional one-hot vector indicating the corresponding amino acid identity, and a binary flag specifying whether the

---

**Algorithm 2** EvoEGF-Mol Sampling

---

1: **function** GeodesicStep($\hat{\mathbf{x}}, \hat{\mathbf{v}}, \hat{\mathbf{b}}, t, \Delta t$)

2:     $\tilde{\sigma}_1(t + \Delta t) \leftarrow \lambda(1 - (t + \Delta t)) + \epsilon$

3:     $\sigma_{t+\Delta t} \leftarrow \left((1 - (t + \Delta t))/\sigma_0^2 + (t + \Delta t)/\tilde{\sigma}_1(t + \Delta t)^2\right)^{-1/2}$

4:     $\boldsymbol{\mu}_{t+\Delta t} \leftarrow \sigma_{t+\Delta t}^2 \left(\frac{(t+\Delta t)\hat{\mathbf{x}}}{\tilde{\sigma}_1(t+\Delta t)^2}\right)$

5:     $\tilde{\boldsymbol{\alpha}}_1^{\mathbf{v}}(t + \Delta t) \leftarrow (1 - \lambda(1 - (t + \Delta t)))\hat{\mathbf{v}} + \lambda(1 - (t + \Delta t))\frac{1}{K_{\mathbf{v}}}\mathbf{1}_{K_{\mathbf{v}}}$

6:     $\boldsymbol{\alpha}_{t+\Delta t}^{\mathbf{v}} \leftarrow (1 - (t + \Delta t))\boldsymbol{\alpha}_0^{\mathbf{v}} + (t + \Delta t)\tilde{\boldsymbol{\alpha}}_1^{\mathbf{v}}(t + \Delta t)$

7:     $\tilde{\boldsymbol{\alpha}}_1^{\mathbf{b}}(t + \Delta t) \leftarrow (1 - \lambda(1 - (t + \Delta t)))\hat{\mathbf{b}} + \lambda(1 - (t + \Delta t))\frac{1}{K_{\mathbf{b}}}\mathbf{1}_{K_{\mathbf{b}}}$

8:     $\boldsymbol{\alpha}_{t+\Delta t}^{\mathbf{b}} \leftarrow (1 - (t + \Delta t))\boldsymbol{\alpha}_0^{\mathbf{b}} + (t + \Delta t)\tilde{\boldsymbol{\alpha}}_1^{\mathbf{b}}(t + \Delta t)$

9:     return $\boldsymbol{\mu}_{t+\Delta t}, \sigma_{t+\Delta t}, \boldsymbol{\alpha}_{t+\Delta t}^{\mathbf{v}}, \boldsymbol{\alpha}_{t+\Delta t}^{\mathbf{b}}$

10: **end function**

**Require:** Network $\boldsymbol{\Phi}$, Protein $P$, Steps $n$, Smoothing coefficient $\lambda$

11: $\boldsymbol{\mu}_0 \leftarrow \mathbf{0}, \sigma_0 \leftarrow 1, \boldsymbol{\alpha}_0^{\mathbf{v}} \leftarrow \mathbf{1}, \boldsymbol{\alpha}_0^{\mathbf{b}} \leftarrow \mathbf{1}, \Delta t \leftarrow 1/n, \lambda \leftarrow 0.2$

12: **for** $i = 0$ to $n - 1$ **do**

13:     $t \leftarrow i/n$

14:     $M_{\mathbf{x}} \sim \mathcal{N}(\boldsymbol{\mu}_t, \sigma_t^2\mathbf{I}), M_{\mathbf{v}} \sim \text{Dir}(\boldsymbol{\alpha}_t^{\mathbf{v}}), M_{\mathbf{b}} \sim \text{Dir}(\boldsymbol{\alpha}_t^{\mathbf{b}})$

15:     $\hat{\mathbf{x}}_M, \hat{\mathbf{v}}_M, \hat{\mathbf{b}}_M \leftarrow \boldsymbol{\Phi}(M_{\mathbf{x}}, M_{\mathbf{v}}, M_{\mathbf{b}}, P, t)$

16:     $\boldsymbol{\mu}_{t+\Delta t}, \sigma_{t+\Delta t}, \boldsymbol{\alpha}_{t+\Delta t}^{\mathbf{v}}, \boldsymbol{\alpha}_{t+\Delta t}^{\mathbf{b}} \leftarrow \text{GeodesicStep}(\hat{\mathbf{x}}_M, \hat{\mathbf{v}}_M, \hat{\mathbf{b}}_M, t, \Delta t)$

17: **end for**

18: Sample $M = (\mathbf{x}_M, \mathbf{v}_M, \mathbf{b}_M)$ from $(\boldsymbol{\mu}_1, \boldsymbol{\alpha}_1^{\mathbf{v}}, \boldsymbol{\alpha}_1^{\mathbf{b}})$

19: return $M$

---

atom belongs to the protein backbone. Ligand atoms are characterized by a one-hot elemental encoding (C, N, O, F, P, S, Cl), augmented with a binary aromaticity indicator. At the edge level, we construct a heterogeneous protein–ligand graph in which edge types are represented by a 4-dimensional one-hot vector denoting ligand–ligand, protein–protein, ligand–protein, and protein–ligand interactions, respectively. Within the ligand subgraph, chemical bonds are encoded using a 4-dimensional one-hot vector corresponding to non-bonded, single, double, and triple bonds; aromatic bonds are converted to their kekulized representations using RDKit. EvoEGF-Mol jointly generates heavy-atom coordinates, atom types, and explicit bond types, requiring no bond assignment; only hydrogen atoms are added afterward, which is straightforward given the predicted bonds.

## B. The Singularity of Dirac Targets on Statistical Manifolds

A critical issue arises when $p_1$ is treated as a Dirac delta distribution $\delta_{\mathbf{x}^*}$ constructed from data samples. Formally, the Dirac distribution can only be approached as a limiting case on the boundary of the statistical manifold $\mathcal{P}$. This singularity leads to a degeneracy in the path construction across both continuous and discrete domains.

Case I: Continuous Variables and Gaussian Collapse

Fig.2 illustrates this situation using a one-dimensional Gaussian distribution as an example. Consider the case where the manifold consists of Gaussian distributions. Let $p_0 = \mathcal{N}(\mu_0, \sigma_0^2)$ and let $p_1$ be a narrow Gaussian $\mathcal{N}(x^*, \sigma_1^2)$ approximating $\delta_{x^*}$. We can see that the precision $\sigma_t^{-2}$ grows linearly with $t/\sigma_1^2$. As we take the limit $\sigma_1 \rightarrow 0$, the rapid growth of precision will cause the variance $\sigma_t^2$ to vanish almost instantaneously. A constant-rate linear change of parameters in the natural parameter space leads to a drastic shift of parameters in the standard parameter space.

Case II: Discrete Simplex and Early Boundary Concentration

The degeneracy observed in the continuous case also manifests in discrete state spaces. Consider a manifold $\mathcal{P}$ of Dirichlet distributions on the $(K - 1)$-simplex $\Delta^{K-1}$. Let $p_1$ denote a degenerate limit of a Dirichlet distribution concentrated at a vertex $\mathbf{v}^*$, obtained as the concentration parameter in the target dimension increases while those of other dimensions remain finite. Under the e-geodesic construction, the natural parameters evolve linearly between a uniform prior $p_0$ with $\alpha_i = 1$ and the degenerate endpoint. This induces a rapid increase in the effective concentration of the target dimension in the early stage of the trajectory. As a result, the distribution becomes highly concentrated near the boundary of the simplex, leading to

a strong imbalance in how probability mass is allocated across time. This imbalance compresses the informative region of the trajectory into a narrow temporal interval, where most meaningful structural transitions occur. Consequently, the model must recover molecular structure from highly corrupted states with limited exposure to intermediate distributions, reducing the effectiveness of parameter learning along the diffusion path.

## C. Comparison of Natural Parameters across Diffusion-based Models

In the context of generative modeling via exponential family density paths, the evolution of natural parameters serves as a critical indicator of learning efficiency and geometric optimality. As illustrated in Fig.7, a fundamental requirement for high-fidelity data synthesis is the divergence of these parameters toward infinity as $t \to 1$. This divergence represents the transition from a diffuse prior to a sharp Dirac delta distribution, effectively "locking" the model's output onto specific data points with zero variance. However, the trajectory by which these parameters reach infinity dictates the numerical stability of the sampling process and the difficulty of the score-matching objective. Traditional diffusion frameworks, such as VP-SDE (DDPM) or BFN, exhibit a "late-stage explosion" where natural parameters remain low value for the majority of the time interval before spiking abruptly near the endpoint. Such high-curvature profiles introduce significant discretization errors during inference, requiring excessively small step sizes to navigate the sudden shift from stochasticity to determinism.

To mitigate these instabilities, our proposed EvoEGF-Mol leverages a dynamic endpoint strategy that redefines the geometric path between noise and data. Unlike the standard EGF, which suffers from a singularity at $t > 0$ when aimed at a fixed Dirac target, EvoEGF regularizes the growth of natural parameters by evolving the target distribution itself over time. As shown by the dash-dotted trajectories in the accompanying figures, EvoEGF achieves a quasi-linear or smooth high-order growth in the natural parameter space. This steady progression ensures that information gain is distributed evenly across the diffusion interval, avoiding the violent numerical stiffness typical of flow-matching or variance-preserving schedules at the terminal boundary.

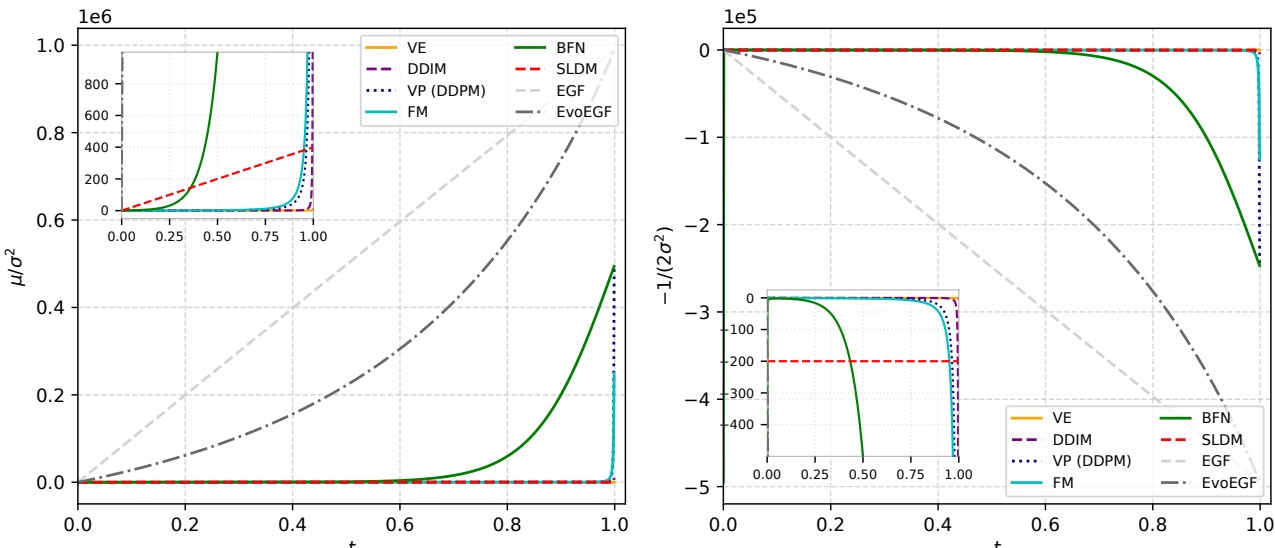

*Figure 7.* Comparison of natural parameters across diffusion-based models for 1D Gaussian distributions. Parameter settings for each model follow Ni et al. (2025).

## D. Dynamic Endpoints are not Equivalent to Non-linear Time Schedules

A natural question is whether the evolving endpoint path in EvoEGF-Mol can be reduced to a fixed endpoint with a non-linear time schedule. Consider

$$\boldsymbol{\eta}_t = (1-t)\boldsymbol{\eta}_0 + t\tilde{\boldsymbol{\eta}}_1(t), \tag{14}$$

and a reparameterized path toward a fixed endpoint

$$\boldsymbol{\eta}_t = (1-s(t))\boldsymbol{\eta}_0 + s(t)\boldsymbol{\eta}_1. \tag{15}$$

For the two paths to be identical for all $t > 0$, we must have

$$\tilde{\boldsymbol{\eta}}_1(t) = \boldsymbol{\eta}_0 + \frac{s(t)}{t}(\boldsymbol{\eta}_1 - \boldsymbol{\eta}_0). \tag{16}$$

Thus, all evolving endpoints $\tilde{\boldsymbol{\eta}}_1(t)$ must lie on the same affine line between $\boldsymbol{\eta}_0$ and $\boldsymbol{\eta}_1$.

This condition is generally not satisfied by EvoEGF-Mol. For a one-dimensional Gaussian endpoint with mean $x^*$ and variance schedule $\tilde{\sigma}_1(t) = \lambda(1-t)$, the direction from $\boldsymbol{\eta}_0$ to $\tilde{\boldsymbol{\eta}}_1(t)$ has slope

$$R(t) = \frac{\tilde{\eta}_{1,2}(t) - \eta_{0,2}}{\tilde{\eta}_{1,1}(t) - \eta_{0,1}} = \frac{2x^*}{-1 + \lambda^2(1-t)^2}, \tag{17}$$

which depends on $t$ when $x^* \neq 0$. Therefore, the endpoint trajectory is not confined to a fixed affine line and cannot be represented by any scalar reparameterization $s(t)$.

This shows that fixed endpoints with non-linear schedules form only a restricted subclass of dynamic-endpoint paths. Dynamic endpoints allow different natural-parameter dimensions to evolve with different relative rates, giving EvoEGF-Mol a more expressive probability-path design.

## E. Connection with Straight-Line Diffusion Model (SLDM)

In this section, we analyze the relationship between our proposed framework and the recently proposed Straight-Line Diffusion Model (SLDM)(Ni et al., 2025). We demonstrate that SLDM can be mathematically derived as a special case of our EGF under specific boundary regularization, providing a geometric interpretation for the efficiency of linear trajectories.

### E.1. Derivation of SLDM from EGF

Recall that the EGF defines a probability path $\rho_t(\cdot|\boldsymbol{\theta}_t)$ connecting a source distribution to a target distribution. The singularity issue in EGF arises when the target is a Dirac distribution (i.e., $\sigma_1 \to 0$), causing the geodesic to exhibit extreme curvature and collapse rapidly towards the boundary of the statistical manifold.

We observe a critical phenomenon: when the boundary conditions of EGF are regularized by setting the initial and target noise levels to a shared small constant, the trajectory degenerates into a linear path. For simplicity of exposition, we illustrate this phenomenon using one-dimensional Gaussian distributions as an example. Specifically, consider the following parameterization in the EGF framework:

Initial State ($t = 0$): $\mu_0 = 0$, $\sigma_0 = \epsilon$, where $\epsilon > 0$ is a small constant.

Target State ($t = 1$): $\mu_1 = x_{\text{data}}$, $\sigma_1 = \epsilon$ (instead of the theoretical target $\sigma_1 \to 0$).

Substituting these parameters into the geodesic equations, the scale parameter $\sigma_t$ remains constant, and the location parameter $\mu_t$ evolves linearly:

$$\sigma_t = \epsilon, \quad \mu_t = t \cdot x_{\text{data}}$$

This corresponds exactly to the noise schedule and trajectory defined in SLDM (Eq. 5 in Ni et al. (2025)), where the diffusion process is formulated as $x_t = (1-t)x_0 + \sigma\epsilon$ (with time reversal).

### E.2. Geometric Interpretation: Transforming Curvature to Flatness

From the perspective of Information Geometry, the manifold of Gaussian distributions equipped with the Fisher-Rao metric has a hyperbolic geometry (essentially the Poincaré half-plane).

The Singularity Case (Standard EGF): A geodesic connecting a standard Gaussian to a Dirac distribution ($\sigma \to 0$) must traverse infinite distance, diving orthogonally towards the boundary. This creates the singularity where $\mu_t$ and $\sigma_t$ change abruptly, leading to large second-order derivatives $\frac{d^2x}{dt^2}$ and high truncation errors in sampling.

The Regularized Case (SLDM): By setting $\sigma_0 = \sigma_1 = \epsilon$, SLDM essentially restricts the transport path to a "horocycle" or a flat submanifold parallel to the boundary. In this regime, the curvature of the manifold vanishes along the path, resulting in a Euclidean straight line.

Therefore, SLDM can be interpreted as a method that bypasses the geometric singularity of reconstructing Dirac distributions by regularizing the target distribution. It trades the exactness of the target (accepting a final noise level $\epsilon$) for the numerical stability of a zero-curvature trajectory ($\frac{d^2 x}{dt^2} = 0$).

### E.3. EvoEGF vs. Static Regularization

This connection highlights the distinct contribution of our EvoEGF:

Static Solution (SLDM): Avoids singularity by fixing the endpoint noise to a non-zero constant $\epsilon$, enforcing a linear path globally. This is efficient but theoretically never recovers the exact clean data distribution (always bounded by $\epsilon$ error).

Dynamic Solution (EvoEGF): Addresses the singularity by evolving the endpoint parameters. This allows the model to approximate the true Dirac distribution ($\sigma \to 0$) more closely while managing the path's curvature dynamically, rather than simplifying the geometry to a flat line ab initio.

## F. Comparison with MolPilot

MolPilot (Qiu et al., 2025) represents a SOTA baseline for SBDD by improving the temporal design of the generative process. Specifically, it decouples modality-specific noise schedules and searches for an optimal off-diagonal schedule via grid discretization and dynamic programming. In contrast, EvoEGF-Mol approaches the same challenge from an information-geometric perspective. Rather than relying on post-hoc schedule optimization, we model molecular states as product exponential-family distributions and define the generative trajectory as an evolving e-geodesic on the Fisher-Rao statistical manifold. This formulation synchronizes continuous coordinates and discrete chemical variables through the natural parameter space, providing a unified geometric path for multi-modal molecular generation.

Although EvoEGF-Mol obtains a slightly lower PB-Valid score than MolPilot, it remains highly competitive while offering a fundamentally different design principle. The proposed dynamic endpoint mechanism mitigates the boundary singularity caused by static Dirac targets and maintains well-conditioned intermediate distributions during generation. Moreover, because the trajectory is defined continuously and analytically, EvoEGF-Mol avoids the additional numerical search required by DP-based optimal scheduling and naturally supports flexible inference-step configurations. We therefore view EvoEGF-Mol as complementary to MolPilot: MolPilot demonstrates the effectiveness of optimized temporal scheduling, whereas EvoEGF-Mol provides a theoretically grounded framework that can serve as a general and extensible foundation for mixed-modality molecular generation.

## G. Loss curve

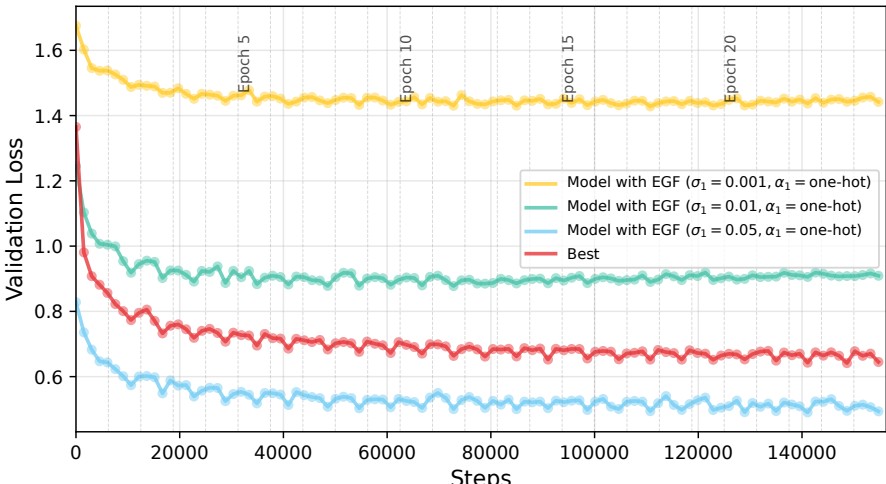

*Figure 8.* Validation loss curves for various experimental configurations.

As shown in Fig. 8, the validation loss curves for EGF reveal a critical trade-off governed by the terminal noise scale ($\sigma_1$).

Our results identify two distinct failure modes: at low scales (e.g., $\sigma_1 = 0.001$ or $\sigma_1 = 0.01$), the model suffers from optimization pathologies where an excessively "sharp" target distribution creates a training singularity. This restricts the effective learning window and causes vanishing gradients across most timesteps. Conversely, higher scales (e.g., $\sigma_1 = 0.05$) lead to trivial convergence; the reconstruction task becomes too simple over the probability path, concentrating the learning signal only at the initial stages and resulting in training imbalance.

These observations suggest that static EGF induces an ill-conditioned objective trapped between unlearnable and trivial regimes. This motivates the transition to EvoEGF. By parameterizing the variance schedule, EvoEGF regularizes the path, smoothing the target singularity and redistributing training pressure more uniformly across the temporal axis for more robust convergence.

## H. Proof

### H.1. Closure of the Exponential Family Under Multiplication on Product Spaces

**Theorem 1.** Under a product factorization, the product of a finite set of probability distributions, each belonging to an exponential family but defined on different domains, is also an exponential-family distribution on the product space.

**1. Definitions and Notation**  Let $\mathbf{x}_k \in \mathcal{X}_k$ denote the random variable for the $k$-th distribution, $k = 1, \dots, K$. The joint random variable is $\mathbf{x} = (\mathbf{x}_1, \dots, \mathbf{x}_K) \in \mathcal{X}_{\text{joint}} \triangleq \mathcal{X}_1 \times \cdots \times \mathcal{X}_K$.

The $k$-th distribution in exponential-family form is:

$$p_k(\mathbf{x}_k | \boldsymbol{\eta}_k) = h_k(\mathbf{x}_k) \exp\left(\boldsymbol{\eta}_k^\top \mathbf{T}_k(\mathbf{x}_k) - A_k(\boldsymbol{\eta}_k)\right), \tag{18}$$

where $\boldsymbol{\eta}_k \in \mathbb{R}^{d_k}$ is the natural parameter vector, $\mathbf{T}_k(\mathbf{x}_k) \in \mathbb{R}^{d_k}$ is the sufficient statistic vector, $A_k(\boldsymbol{\eta}_k)$ is the log-partition function, and $h_k(\mathbf{x}_k)$ is the base measure.

**2. Product Distribution on the Joint Space**  We define the product distribution on the joint space $\mathcal{X}_{\text{joint}}$ as:

$$f(\mathbf{x}_1, \dots, \mathbf{x}_K) = \prod_{k=1}^K p_k(\mathbf{x}_k | \boldsymbol{\eta}_k). \tag{19}$$

Substituting the exponential-family form:

$$f(\mathbf{x}_1, \dots, \mathbf{x}_K) = \prod_{k=1}^K \left[ h_k(\mathbf{x}_k) \exp\left(\boldsymbol{\eta}_k^\top \mathbf{T}_k(\mathbf{x}_k) - A_k(\boldsymbol{\eta}_k)\right) \right] \tag{20}$$

$$= \left(\prod_{k=1}^K h_k(\mathbf{x}_k)\right) \exp\left(\sum_{k=1}^K \boldsymbol{\eta}_k^\top \mathbf{T}_k(\mathbf{x}_k) - \sum_{k=1}^K A_k(\boldsymbol{\eta}_k)\right). \tag{21}$$

To combine all terms into a single exponential-family form, we define:

$$\mathbf{T}(\mathbf{x}_1, \dots, \mathbf{x}_K) \triangleq \begin{bmatrix} \mathbf{T}_1(\mathbf{x}_1) \\ \vdots \\ \mathbf{T}_K(\mathbf{x}_K) \end{bmatrix}, \qquad \boldsymbol{\eta} \triangleq \begin{bmatrix} \boldsymbol{\eta}_1 \\ \vdots \\ \boldsymbol{\eta}_K \end{bmatrix}, \qquad h(\mathbf{x}_1, \dots, \mathbf{x}_K) \triangleq \prod_{k=1}^K h_k(\mathbf{x}_k), \qquad C \triangleq \sum_{k=1}^K A_k(\boldsymbol{\eta}_k). \tag{22}$$

Then the product can be written as:

$$f(\mathbf{x}_1, \dots, \mathbf{x}_K) = h(\mathbf{x}_1, \dots, \mathbf{x}_K) \exp\left((\boldsymbol{\eta})^\top \mathbf{T}(\mathbf{x}_1, \dots, \mathbf{x}_K) - C\right). \tag{23}$$

**Summary**  The resulting product distribution $p(\mathbf{x}_1, \dots, \mathbf{x}_K)$ is an exponential-family distribution on the product space $\mathcal{X}_1 \times \cdots \times \mathcal{X}_K$ with:

- **Natural parameter:** $\boldsymbol{\eta} = [\boldsymbol{\eta}_1, \dots, \boldsymbol{\eta}_K]^\top$,

- **Sufficient statistic:** $\mathbf{T}(\mathbf{x}_1, \dots, \mathbf{x}_K) = [\mathbf{T}_1(\mathbf{x}_1), \dots, \mathbf{T}_K(\mathbf{x}_K)]^\top$,

- **Base measure:** $h(\mathbf{x}_1, \dots, \mathbf{x}_K) = \prod_{k=1}^K h_k(\mathbf{x}_k)$.

## H.2. Derivation of the Induced Sample-Space Velocity Field

This part provides an optional continuous-flow interpretation of the density path defined in the natural-parameter space. It is not the training mechanism of EvoEGF-Mol, which directly optimizes endpoint-induced one-step distributions as described in Sec. 3.3.

Let

$$p_t(\mathbf{x}) = h(\mathbf{x}) \exp\{\boldsymbol{\eta}_t^\top \mathbf{T}(\mathbf{x}) - A(\boldsymbol{\eta}_t)\} \tag{24}$$

be a smooth exponential-family path. Denote

$$\boldsymbol{\mu}_t = \nabla A(\boldsymbol{\eta}_t) = \mathbb{E}_{p_t}[\mathbf{T}(\mathbf{X})], \qquad s_t(\mathbf{x}) = \big(\mathbf{T}(\mathbf{x}) - \boldsymbol{\mu}_t\big)^\top \dot{\boldsymbol{\eta}}_t. \tag{25}$$

Then

$$\frac{\partial \log p_t(\mathbf{x})}{\partial t} = s_t(\mathbf{x}), \qquad \frac{\partial p_t(\mathbf{x})}{\partial t} = p_t(\mathbf{x}) s_t(\mathbf{x}). \tag{26}$$

Since $\mathbb{E}_{p_t}[s_t(\mathbf{X})] = 0$, this derivative integrates to zero and therefore preserves total probability.

When the sample space is continuous and we want to realize the same density path through a velocity field $\mathbf{u}_t(\mathbf{x})$, the field must satisfy the continuity equation

$$\frac{\partial p_t(\mathbf{x})}{\partial t} + \nabla_{\mathbf{x}} \cdot \big(p_t(\mathbf{x}) \mathbf{u}_t(\mathbf{x})\big) = 0, \tag{27}$$

or equivalently

$$\nabla_{\mathbf{x}} \cdot \big(p_t(\mathbf{x}) \mathbf{u}_t(\mathbf{x})\big) = -p_t(\mathbf{x})\big(\mathbf{T}(\mathbf{x}) - \nabla A(\boldsymbol{\eta}_t)\big)^\top \dot{\boldsymbol{\eta}}_t. \tag{28}$$

Eq. 28 is a weighted divergence equation. Its solution is generally not unique: if $\mathbf{u}_t$ is one solution and $\mathbf{w}_t$ satisfies $\nabla_{\mathbf{x}} \cdot (p_t \mathbf{w}_t) = 0$, then $\mathbf{u}_t + \mathbf{w}_t$ is another solution. Thus, the natural-parameter velocity $\dot{\boldsymbol{\eta}}_t$ does not determine a unique sample-space vector field.

A canonical representative can be selected by minimizing kinetic energy in $L^2(p_t)$. The minimizer is the gradient field

$$\mathbf{u}_t^\star(\mathbf{x}) = \nabla_{\mathbf{x}} \phi_t(\mathbf{x}), \tag{29}$$

where $\phi_t$ solves the weighted Poisson equation

$$-\nabla_{\mathbf{x}} \cdot \big(p_t(\mathbf{x}) \nabla_{\mathbf{x}} \phi_t(\mathbf{x})\big) = p_t(\mathbf{x})\big(\mathbf{T}(\mathbf{x}) - \boldsymbol{\mu}_t\big)^\top \dot{\boldsymbol{\eta}}_t, \tag{30}$$

with appropriate decay or no-flux boundary conditions. For relaxed continuous sample spaces such as the interior of a simplex for Dirichlet variables, one may similarly write the equation on a Riemannian manifold $(\mathcal{M}, g)$, the corresponding expression is

$$\mathbf{u}_t^\star = \mathrm{grad}_g \phi_t, \qquad -\mathrm{div}_g\big(p_t \, \mathrm{grad}_g \phi_t\big) = p_t\big(\mathbf{T} - \boldsymbol{\mu}_t\big)^\top \dot{\boldsymbol{\eta}}_t. \tag{31}$$

In one dimension, under the usual vanishing-flux boundary condition at $-\infty$, Eq. 28 gives

$$u_t(x) = -\frac{1}{p_t(x)} \int_{-\infty}^x p_t(y)\big(\mathbf{T}(y) - \boldsymbol{\mu}_t\big)^\top \dot{\boldsymbol{\eta}}_t \, dy. \tag{32}$$

## H.3. Local Fisher-Rao Interpretation of the KL Objective

We now state the precise infinitesimal relation between the local KL transport cost used by EvoEGF and flow matching on the natural-parameter manifold $(\Omega, \mathbf{G})$.

For a regular exponential family, the Fisher information matrix in natural coordinates is

$$\mathbf{G}(\boldsymbol{\eta}) = \mathbb{E}_{p_{\boldsymbol{\eta}}}\Big[\big(\mathbf{T}(\mathbf{X}) - \nabla A(\boldsymbol{\eta})\big)\big(\mathbf{T}(\mathbf{X}) - \nabla A(\boldsymbol{\eta})\big)^\top\Big] = \nabla^2 A(\boldsymbol{\eta}). \tag{33}$$

The KL divergence between two nearby natural parameters satisfies the second-order expansion

$$D_{\mathrm{KL}}\big(p_{\boldsymbol{\eta}+\boldsymbol{\xi}_1} \,\|\, p_{\boldsymbol{\eta}+\boldsymbol{\xi}_2}\big) = \frac{1}{2}(\boldsymbol{\xi}_1 - \boldsymbol{\xi}_2)^\top \mathbf{G}(\boldsymbol{\eta})(\boldsymbol{\xi}_1 - \boldsymbol{\xi}_2) + o(\|\boldsymbol{\xi}_1\|^2 + \|\boldsymbol{\xi}_2\|^2). \tag{34}$$

Let the true one-step parameter update and the model one-step update be

$$\boldsymbol{\eta}_t^+ = \boldsymbol{\eta}_t + \mathbf{v}_t \Delta t, \qquad \hat{\boldsymbol{\eta}}_t^+ = \boldsymbol{\eta}_t + \hat{\mathbf{v}}_t \Delta t, \tag{35}$$

where $\mathbf{v}_t = \dot{\boldsymbol{\eta}}_t$ and $\hat{\mathbf{v}}_t \in T_{\boldsymbol{\eta}_t}\Omega$ is the model-predicted parameter velocity. Substituting $\boldsymbol{\xi}_1 = \mathbf{v}_t \Delta t$ and $\boldsymbol{\xi}_2 = \hat{\mathbf{v}}_t \Delta t$ into Eq. 34 yields

$$D_{\mathrm{KL}}\left(p_{\boldsymbol{\eta}_t^+} \,\|\, p_{\hat{\boldsymbol{\eta}}_t^+}\right) = \frac{\Delta t^2}{2}(\mathbf{v}_t - \hat{\mathbf{v}}_t)^\top \mathbf{G}(\boldsymbol{\eta}_t)(\mathbf{v}_t - \hat{\mathbf{v}}_t) + o(\Delta t^2). \tag{36}$$

Therefore

$$\lim_{\Delta t \to 0} \frac{1}{\Delta t^2} D_{\mathrm{KL}}\left(p_{\boldsymbol{\eta}_t^+} \,\|\, p_{\hat{\boldsymbol{\eta}}_t^+}\right) = \frac{1}{2}\|\mathbf{v}_t - \hat{\mathbf{v}}_t\|^2_{\mathbf{G}(\boldsymbol{\eta}_t)}. \tag{37}$$

This establishes that, to second order, the one-step KL loss penalizes the mismatch between true and model-induced parameter displacements in the Fisher-Rao norm. For the evolving endpoint path

$$\boldsymbol{\eta}_t = (1-t)\boldsymbol{\eta}_0 + t\tilde{\boldsymbol{\eta}}_1(t), \tag{38}$$

the target parameter velocity is

$$\mathbf{v}_t = \dot{\boldsymbol{\eta}}_t = \tilde{\boldsymbol{\eta}}_1(t) - \boldsymbol{\eta}_0 + t\dot{\tilde{\boldsymbol{\eta}}}_1(t). \tag{39}$$

In EvoEGF-Mol, the network predicts a terminal parameter $\hat{\boldsymbol{\eta}}_1(t)$ rather than an explicit velocity. The corresponding one-step prediction is obtained by constructing the model-induced next parameter $\widehat{\boldsymbol{\eta}}_{t+\Delta t}$ from this endpoint prediction and comparing it with $\boldsymbol{\eta}_{t+\Delta t}$ through KL. If one wants to express this comparison in velocity notation, an induced predicted velocity may be defined, for example, as

$$\hat{\mathbf{v}}_t = \hat{\boldsymbol{\eta}}_1(t) - \boldsymbol{\eta}_0 + t\,\partial_t \hat{\boldsymbol{\eta}}_1(t), \tag{40}$$

or more directly by the finite-difference update $(\widehat{\boldsymbol{\eta}}_{t+\Delta t} - \boldsymbol{\eta}_t)/\Delta t$. Thus, the EvoEGF KL objective admits a local Fisher-Rao flow-matching interpretation in the exponential-family parameter manifold, while the sample-space velocity characterization remains only the weighted-divergence problem in Appendix H.2.

## I. Derivation of the Loss Function

We derive the loss function $\mathcal{L}_{\mathbf{x},t}$ by minimizing the Kullback–Leibler (KL) divergence between the true intermediate distribution $p_{\mathbf{x},t} = \mathcal{N}(\boldsymbol{\mu}_t, \sigma_t^2 \mathbf{I})$ and the predicted distribution $\hat{p}_{\mathbf{x},t} = \mathcal{N}(\hat{\boldsymbol{\mu}}_t, \sigma_t^2 \mathbf{I})$.

1. KL Divergence for Isotropic Gaussians

For two multivariate Gaussian distributions with identical isotropic covariance $\boldsymbol{\Sigma} = \sigma_t^2 \mathbf{I}$, the KL divergence simplifies to the squared Euclidean distance scaled by the variance(Soch et al., 2024):

$$\begin{aligned}
D_{\mathrm{KL}}(p_{\mathbf{x},t} \,\|\, \hat{p}_{\mathbf{x},t}) &= \frac{1}{2}\left[\mathrm{tr}(\boldsymbol{\Sigma}^{-1}\boldsymbol{\Sigma}) - d + (\boldsymbol{\mu}_t - \hat{\boldsymbol{\mu}}_t)^\top \boldsymbol{\Sigma}^{-1}(\boldsymbol{\mu}_t - \hat{\boldsymbol{\mu}}_t) + \ln\frac{|\boldsymbol{\Sigma}|}{|\boldsymbol{\Sigma}|}\right] \\
&= \frac{1}{2}\left[d - d + \frac{1}{\sigma_t^2}\|\boldsymbol{\mu}_t - \hat{\boldsymbol{\mu}}_t\|^2 + 0\right] \\
&= \frac{1}{2\sigma_t^2}\|\boldsymbol{\mu}_t - \hat{\boldsymbol{\mu}}_t\|^2
\end{aligned} \tag{41}$$

2. Substitution of Exponential Geodesic Equations

Recall the mean schedule for the e-geodesic:

$$\boldsymbol{\mu}_t = \sigma_t^2 \left[\frac{(1-t)\boldsymbol{\mu}_0}{\sigma_0^2} + \frac{t\boldsymbol{\mu}_1}{\sigma_1^2}\right] \tag{42}$$

The estimated mean $\hat{\boldsymbol{\mu}}_t$ is defined similarly using the estimated target $\hat{\boldsymbol{\mu}}_1$:

$$\hat{\boldsymbol{\mu}}_t = \sigma_t^2 \left[\frac{(1-t)\boldsymbol{\mu}_0}{\sigma_0^2} + \frac{t\hat{\boldsymbol{\mu}}_1}{\sigma_1^2}\right] \tag{43}$$

Subtracting $\hat{\boldsymbol{\mu}}_t$ from $\boldsymbol{\mu}_t$, the terms involving $\boldsymbol{\mu}_0$ cancel out:

$$\boldsymbol{\mu}_t - \hat{\boldsymbol{\mu}}_t = \sigma_t^2 \left( \frac{t\boldsymbol{\mu}_1}{\sigma_1^2} - \frac{t\hat{\boldsymbol{\mu}}_1}{\sigma_1^2} \right) = \frac{t\sigma_t^2}{\sigma_1^2}(\boldsymbol{\mu}_1 - \hat{\boldsymbol{\mu}}_1) \tag{44}$$

3. Final Loss Function Formulation

Substituting the difference derived above into the KL divergence equation:

$$
\begin{aligned}
\mathcal{L}_{\mathbf{x},t} = D_{\mathrm{KL}}(p_{\mathbf{x},t} \parallel \hat{p}_{\mathbf{x},t}) &= \frac{1}{2\sigma_t^2} \left\| \frac{t\sigma_t^2}{\sigma_1^2}(\boldsymbol{\mu}_1 - \hat{\boldsymbol{\mu}}_1) \right\|^2 \\
&= \frac{1}{2\sigma_t^2} \cdot \frac{t^2\sigma_t^4}{\sigma_1^4} \|\boldsymbol{\mu}_1 - \hat{\boldsymbol{\mu}}_1\|^2 \\
&= \frac{t^2\sigma_t^2}{2\sigma_1^4} \|\boldsymbol{\mu}_1 - \hat{\boldsymbol{\mu}}_1\|^2
\end{aligned}
\tag{45}
$$

Taking the expectation over the data, we obtain the final objective:

$$\mathcal{L}_{\mathbf{x},t} = \mathbb{E}_{p_{\mathbf{x}}} \left[ \frac{t^2\sigma_t^2 \|\boldsymbol{\mu}_1 - \hat{\boldsymbol{\mu}}_1\|^2}{2\sigma_1^4} \right] \tag{46}$$

Following the framework used for Gaussian e-geodesics, we derive the loss function for discrete variables modeled via the Dirichlet distribution. We minimize the KL divergence between the true intermediate distribution $p_{\mathbf{v},t} = \mathrm{Dir}(\boldsymbol{\alpha}_t)$ and the predicted distribution $\hat{p}_{\mathbf{v},t} = \mathrm{Dir}(\hat{\boldsymbol{\alpha}}_t)$. The same derivation applies to bond types.

The ratio of the normalization constants is expressed as(Soch et al., 2024):

$$\ln \frac{B(\boldsymbol{\alpha}_t)}{B(\hat{\boldsymbol{\alpha}}_t)} = \ln \frac{\Gamma(\sum_j \hat{\alpha}_{t,j})}{\Gamma(\sum_j \alpha_{t,j})} + \sum_{i=1}^{K} \ln \frac{\Gamma(\alpha_{t,i})}{\Gamma(\hat{\alpha}_{t,i})} \tag{47}$$

The loss $\mathcal{L}_{\mathbf{v},t}$ is the expectation over the data and sampled time $t$:

$$\mathcal{L}_{\mathbf{v},t} = D_{\mathrm{KL}}(p_{\mathbf{v},t} \| \hat{p}_{\mathbf{v},t}) = \mathbb{E}_{p_{\mathbf{v}}} \left[ \ln \frac{B(\boldsymbol{\alpha}_t)}{B(\hat{\boldsymbol{\alpha}}_t)} + \sum_{i=1}^{K} (\hat{\alpha}_{t,i} - \alpha_{t,i}) \left( \psi(\hat{\alpha}_{t,i}) - \psi\left( \sum_{j=1}^{K} \hat{\alpha}_{t,j} \right) \right) \right] \tag{48}$$

## J. More Evaluation Results

### J.1. Performance on Active Molecules Discovery

#### J.1.1. MolGenBench Metrics

In this section, we provide formal definitions and descriptions of the metrics used to evaluate the molecular generative models within the MolGenBench framework.

1. Pass Rate (ChemFilter)

The Pass Rate evaluates the chemical quality and medicinal chemistry relevance of the generated molecules. It measures the proportion of molecules that survive a suite of industry-standard filters designed to exclude unstable, reactive, or promiscuous "high-risk" structures (e.g., Eli Lilly reactivity rules and NIBR criteria, and ChEMBL rules(Schuffenhauer et al., 2020; Bruns & Watson, 2012; Gaulton et al., 2012)).

$$\text{Pass Rate} = \frac{|\mathcal{M}_{\text{passed}}|}{|\mathcal{M}_{\text{total}}|}$$

where $\mathcal{M}_{\text{total}}$ is the set of all generated molecules and $\mathcal{M}_{\text{passed}}$ is the subset of molecules that satisfy all chemical filters.

2. Hit Recovery

Hit Recovery measures the model's ability to successfully reconstruct or "find" known active molecules or Bemis-Murcko scaffolds(Bemis & Murcko, 1996) for a specific target. This metric is used to assess the baseline generative performance across various target categories, such as proteins included in or excluded from the training distribution (e.g., CrossDock).

3. Hit Rate

The Hit Rate quantifies the efficiency of the generative model by measuring how many of the generated samples are actual active entities. Unlike computational proxies, this metric uses a reference library of known actives to provide an unbiased assessment of discovery efficiency.

$$\text{Hit Rate}_{type} = \frac{M_{\text{active}}}{M_{\text{sampled}}}, \quad type \in \{\text{SMILES, Scaffold}\}$$

where $M_{\text{active}}$ denotes the number of unique active molecules (or molecules containing active scaffolds) identified in the sampled set, and $M_{\text{sampled}}$ is the total number of molecules generated for evaluation.

4. Hit Fraction

The Hit Fraction measures the coverage of the active chemical space. While the Hit Rate focuses on efficiency, the Hit Fraction evaluates the extent to which a model can recover the diversity of known ligands for a specific target.

$$\text{Hit Fraction}_{type} = \frac{F_{\text{active}}}{F_{\text{ref}}}, \quad type \in \{\text{SMILES, Scaffold}\}$$

where $F_{\text{active}}$ is the number of unique active molecules (or scaffolds) recovered by the model, and $F_{\text{ref}}$ is the total number of unique reference active molecules (or scaffolds) known for that specific target.

5. Target-Aware Score (TAScore)

The TAScore assesses the specificity of the model by comparing the success rate when conditioned on a specific target against the background success rate across all targets. A high TAScore indicates that the model's output is genuinely driven by target-specific information rather than a general bias toward common active-like molecules.

$$\text{TAScore}_{type,i} = \frac{S_i/S_{\text{total}}}{R_i/R_{\text{total}}}, \quad type \in \{\text{SMILES, Scaffold}\}$$

For a target $i$:

$S_{\text{total}}$: Total distinct molecules generated when conditioned on target $i$.

$S_i$: Subset of $S_{\text{total}}$ that matches known actives for target $i$.

$R_{\text{total}}$: Total distinct molecules generated by the model across all targets in the benchmark.

$R_i$: Subset of $R_{\text{total}}$ that matches known actives for target $i$ (representing the background hit rate).

J.1.2. RESULTS

Our proposed method consistently outperforms SBDD baselines in active molecule rediscovery. As shown in table 3, On the "In-distribution" set, we achieve a Pass Rate of 37.52% and recover 7 hits, while early diffusion-based models fail to rediscover any known actives (Hit Recovery = 0). Despite the relative gains, the absolute performance reveals a substantial gap between current generative capabilities and real-world drug discovery requirements. Even our top-performing model yields a Hit Fraction of only 0.01%. This near-zero absolute success rate underscores the extreme difficulty of de novo rediscovery using purely structure-based cues. For proteins not present in the training set (Not), the Hit Recovery for our model drops to 0, illustrating that current SBDD models struggle to generalize their learned geometric priors to novel biological targets.

These results suggest that optimizing for geometric complementarity alone (e.g., via CrossDock) is insufficient for capturing pharmacophoric requirements. Future breakthroughs likely necessitate training on explicit bioactivity datasets to bridge the gap from "physically plausible" to "biochemically active" molecular generation.

*Table 3.* Performance on MolGenBench regarding molecules metrics. Definitions: **In** (Proteins in CrossDock), **In(RM.)** (Proteins in CrossDock(remove SMILES in CrossDock train set)), **Not** (Proteins not in CrossDock). (↑) / (↓) indicates larger / smaller is better. The top-2 results are **bolded** and underlined. Baselines are evaluated based on released samples from Cao et al. (2025).

| Methods | Pass Rate In (%) | Pass Rate Not (%) | Hit Recovery (↑) In | Hit Recovery (↑) In(RM.) | Hit Recovery (↑) Not | Hit Rate (↑) In (%) | Hit Rate (↑) In(RM.) (%) | Hit Rate (↑) Not (%) | Hit Fraction (↑) In (%) | Hit Fraction (↑) In(RM.) (%) | Hit Fraction (↑) Not (%) | TAScore Count In 0-1 | In 1-10 | In 10-100 | In >100 | Not 0-1 | Not 1-10 | Not 10-100 | Not >100 |
|---|---|---|---|---|---|---|---|---|---|---|---|---|---|---|---|---|---|---|---|
| Pocket2Mol | 20.09 | 17.96 | 0 | 0 | 0 | 0.00 | 0.00 | 0.00 | 0.00 | 0.00 | 0.00 | 85 | 0 | 0 | 0 | 35 | 0 | 0 | 0 |
| TargetDiff | 5.99 | 6.64 | 0 | 0 | 0 | 0.00 | 0.00 | 0.00 | 0.00 | 0.00 | 0.00 | 85 | 0 | 0 | 0 | 35 | 0 | 0 | 0 |
| DecompDiff | 22.90 | 20.77 | 2 | 2 | 0 | 0.00 | 0.00 | 0.00 | 0.00 | 0.00 | 0.00 | 80 | 0 | 2 | 0 | 30 | 0 | 0 | 0 |
| MolCRAFT | 25.93 | 26.35 | 3 | 3 | 1 | 0.01 | 0.01 | 0.00 | 0.00 | 0.00 | 0.00 | 82 | 0 | 1 | 2 | 34 | 0 | 1 | 0 |
| EvoEGF-Mol | **37.52** | **33.75** | **7** | **5** | 0 | **0.03** | **0.03** | 0.00 | **0.01** | **0.01** | 0.00 | 78 | 1 | 4 | 2 | 35 | 0 | 0 | 0 |

## J.2. Full Ablation Studies

To validate the design choices in EvoEGF-Mol, we conducted a comprehensive ablation study analyzing the impact of the evolving strategy, Dirichlet parameterization, smoothing coefficients, and explicit bond generation. The results are summarized in Table 4.Since there are no significant differences between the models in terms of Connectivity, QED, SA, and Clash Ratio, we focus on PoseBuster Validity (PB-Valid), Binding Affinity (Vina Score, Vina Min, Vina Dock), and Strain Energy (SE).

**Effect of Evolving E-Geodesic mechanism.** We first evaluate the proposed "evolving" endpoint strategy against a Naïve Exponential Geodesic baseline, where the target distribution is modeled as a fixed Dirac delta function—approximated via a vanishingly small Gaussian standard deviation and near-one-hot concentration parameters—throughout the trajectory. Experimental results indicate that the Naïve approach fails catastrophically: it achieves a PoseBuster Validity (PB-Valid) of only $64.12\%$, exhibits high SE percentiles, and yields positive Vina scores ($+1.67$) indicative of significant repulsive forces. This failure confirms that fixing the target variance to zero or the concentration to a one-hot state prematurely induces excessively imbalanced training pressure, thereby hindering effective convergence. In contrast, our method dynamically evolves the variance and concentration parameters ($\tilde{\sigma}_1(t)$ and $\tilde{\boldsymbol{\alpha}}_1(t)$), leading to substantial improvements in generation quality. This underscores that the evolving strategy is essential for e-geodesic-based generative paths on the Fisher-Rao manifold within our framework.

**Effect of Dirichlet Distribution Parameter Settings.** The configuration of the target atom-type distribution $p_{\mathbf{v},1}(\mathbf{v})$ significantly influences generative performance. Comparing our approach to standard Dirichlet Flow Matching (where $\boldsymbol{\alpha}_0 = 1$ and $\boldsymbol{\alpha}_1$ is a vector with the target category set to infinity) reveals that aggressive concentration toward the target category early in the trajectory hinders exploration. This is evidenced by the highest SE first and second quartiles ($20.33$ and $49.65$, respectively) and a reduced validity of $88.32\%$. While adopting a uniform concentration prior ($\boldsymbol{\alpha}_0 = \frac{1}{K}\mathbf{1}_K \to \boldsymbol{\alpha}_1 =$ nearly one-hot) improves results relative to standard Dirichlet Flow Matching, it remains suboptimal compared to our proposed strategy.

**Effect of Smoothing Coefficients and Vectors.** Analysis of various smoothing coefficients reveals a "sweet spot" at $0.2$. A lower coefficient ($0.1$) significantly degrades the binding affinity of generated molecules (Vina Score $-4.72$), whereas higher coefficients ($0.3, 0.4$) yield marginally inferior performance across multiple metrics compared to our best configuration. Furthermore, employing a smoothing vector $\boldsymbol{\alpha}_0 = 1$ most closely approximates our optimal setup—which mixes a one-hot concentration parameter with a uniform one. These findings suggest that our specific evolving schedule strikes the ideal balance between categorical determination and generative quality. Notably, the smoothing coefficient and vector do not require per-protein tuning, as they primarily smooth the information-geometric path rather than encode target-specific information. By mitigating pathological curvature on the statistical manifold, the resulting smoothing scheme generalizes effectively across diverse protein pockets.

**Effect of Bond Diffusion.** Finally, we assess the necessity of explicit bond generation by comparing our full model to a "Without Bond Generation" variant. In this ablation, we retrained the model without the bond generation objective; the model generates only atomic coordinates and types, with the final molecular structure assembled via OpenBabel post-hoc. While this variant achieves a competitive Vina Score ($-6.25$) by effectively positioning atoms within binding pockets, its structural integrity is significantly compromised. Specifically, it exhibits higher SE quartiles and lower PB-Valid ($90.82\%$) compared to the full model. This indicates that while coordinate flows can successfully identify favorable binding regions, the explicit bond generation module is indispensable for ensuring that internal geometries respect chemical valency and bond length constraints, thereby maintaining overall molecular validity.

*Table 4.* Ablation studies on CrossDock. (↑) / (↓) indicates larger / smaller is better. The top-2 results are **bolded** and underlined. Note: CR is short for Clash Ratio.

| Methods | PB-Valid Avg. (↑) | Vina Score (↓) | | Vina Min (↓) | | Vina Dock (↓) | | Strain Energy (↓) | | | Connected Avg. (↑) | QED Avg. (↑) | SA Avg. (↑) | CR Avg. (↓) |
|---|---|---|---|---|---|---|---|---|---|---|---|---|---|---|
| | | Avg. | Med. | Avg. | Med. | Avg. | Med. | 25% | 50% | 75% | | | | |
| Dirichlet $\alpha_0 = \frac{1}{K}$ | 90.9% | -5.86 | -6.71 | -6.88 | -6.99 | -7.45 | -7.73 | 11.73 | 32.66 | 71.61 | 97.8% | **0.55** | 0.73 | 0.27 |
| Dirichlet FM alpha setting | 88.3% | -5.56 | -6.39 | -6.48 | -6.56 | -7.18 | -7.35 | 20.33 | 49.65 | 110.72 | 94.3% | 0.52 | 0.68 | **0.22** |
| Dirichlet smooth vector=$\alpha_0$ | 92.5% | -5.95 | -6.78 | -6.91 | -7.03 | -7.58 | -7.80 | 9.62 | 27.98 | 60.95 | 97.3% | 0.53 | 0.74 | 0.24 |
| Smooth coefficient=0.1 | 88.4% | -4.72 | -6.60 | -6.23 | -6.91 | -7.32 | -7.79 | 9.60 | 27.59 | 62.92 | 98.0% | 0.54 | 0.73 | 0.29 |
| Smooth coefficient=0.3 | 92.9% | -6.14 | -6.80 | -7.03 | -7.04 | **-7.81** | -7.84 | 10.48 | 30.03 | 64.46 | 97.4% | 0.53 | 0.74 | 0.24 |
| Smooth coefficient=0.4 | 92.3% | **-6.27** | -6.80 | -7.04 | -7.08 | -7.72 | -7.86 | 12.26 | 33.66 | 72.93 | 97.2% | 0.53 | 0.74 | 0.26 |
| Modeling without bond | 90.8% | -6.25 | -6.83 | **-7.10** | -7.11 | **-7.81** | -7.89 | 12.40 | 34.48 | 79.52 | 97.2% | 0.54 | 0.74 | **0.22** |
| Model with EGF ($\sigma_1 = 0.05$) | 64.1% | 1.67 | -5.96 | -2.47 | -6.28 | -3.37 | -7.21 | 12.67 | 46.21 | 176.03 | 97.0% | 0.51 | 0.60 | 0.43 |
| Sampling steps=20 | 85.8% | -5.50 | -6.42 | -6.57 | -6.76 | -7.53 | -7.58 | 18.48 | 53.74 | 129.70 | 97.4% | 0.53 | 0.67 | 0.30 |
| Sampling steps=50 | 92.0% | -5.97 | -6.76 | -6.92 | -7.04 | -7.74 | -7.81 | 11.67 | 32.93 | 71.72 | 98.4% | 0.53 | 0.73 | 0.25 |
| Best | **93.4%** | -6.14 | **-6.89** | -6.98 | **-7.12** | -7.72 | -7.88 | 8.94 | 25.96 | 56.65 | **98.6%** | 0.53 | **0.75** | 0.24 |

## J.3. Full PoseBusters Results

We evaluate the physical and chemical integrity of molecules generated by EvoEGF-Mol using PoseBusters. As shown in Table 5, EvoEGF-Mol demonstrates exceptional performance across all 20 biophysical tests. Notably, the model achieves a 100% success rate in chemical validity (sanitization and InChI conversion) and near-perfect scores in local geometry, with 99.9% passing bond length and angle checks.

Unlike many 3D generative models that produce high-energy or clashing structures, EvoEGF-Mol yields physically plausible poses, with 98.97% avoiding internal steric clashes and 97.87% satisfying internal energy constraints. Furthermore, the model respects the binding site's boundaries, evidenced by the 99.02% pass rate for protein volume overlap. These results indicate that EvoEGF-Mol effectively learns the underlying physical constraints of protein-ligand systems.

*Table 5.* Percentage of molecules generated by EvoEGF-Mol that have passed the PoseBusters validity checks on CrossDock test set.

| Indicator | Percentage of molecules passing the check |
|---|---|
| Mol pred loaded | 100.00% |
| Mol cond loaded | 100.00% |
| Sanitization | 100.00% |
| Inchi convertible | 100.00% |
| All atoms connected | 98.60% |
| Bond lengths | 99.99% |
| Bond angles | 99.98% |
| Internal steric clash | 98.97% |
| Aromatic ring flatness | 100.00% |
| Double bond flatness | 99.80% |
| Internal energy | 97.87% |
| Protein-ligand maximum distance | 100.00% |
| Minimum distance to protein | 97.69% |
| Minimum distance to organic cofactors | 99.95% |
| Minimum distance to inorganic cofactors | 99.72% |
| Minimum distance to waters | 99.95% |
| Volume overlap with protein | 99.02% |
| Volume overlap with organic cofactors | 100.00% |
| Volume overlap with inorganic cofactors | 100.00% |
| Volume overlap with waters | 100.00% |

## J.4. Intermediate States Analysis

By evaluating Validity (successful RDKit parsing into SMILES) and Completeness (parsing into a single, unified molecule) across early timesteps, we observe that EvoEGF-Mol accelerates the emergence of "molecular prototypes" compared to baselines with separately refined coordinate and topology paths (Table 6). Remarkably, the geometric fidelity of our structures

at $t = 0.6$ is already superior to the fully converged outputs of TargetDiff and remains competitive with MolCRAFT (Table 7). This indicates that EvoEGF-Mol captures essential chemical constraints well before the completion of the diffusion process.

*Table 6.* Comparison of intermediate states regarding validity and completeness.

| Model | Validity ($\uparrow$) | | | | Completeness ($\uparrow$) | | | |
|---|---|---|---|---|---|---|---|---|
| | $t=0.3$ | $t=0.4$ | $t=0.5$ | $t=0.6$ | $t=0.3$ | $t=0.4$ | $t=0.5$ | $t=0.6$ |
| TargetDiff | 0.29 | 0.36 | 0.44 | 0.56 | 0.09 | 0.11 | 0.19 | 0.30 |
| MolCRAFT | 0.78 | 0.90 | 0.96 | **0.99** | 0.37 | 0.60 | 0.86 | 0.92 |
| EvoEGF-Mol | **0.96** | **0.97** | **0.98** | **0.99** | **0.39** | **0.75** | **0.94** | **0.97** |

*Table 7.* Comparison of intermediate states regarding bond length and bond angle distributions.

| Model | $JS_{BL}$ ($\downarrow$) | $JS_{BA}$ ($\downarrow$) |
|---|---|---|
| TargetDiff ($t = 1.0$) | 0.33 | 0.39 |
| MolCRAFT ($t = 1.0$) | **0.31** | **0.33** |
| EvoEGF-Mol ($t = 0.6$) | **0.31** | 0.34 |

## J.5. Comformation Stability

To evaluate the 2D structural fidelity of generated molecules across atom types, ring sizes, and functional groups, two metrics are employed in CBGBench(Lin et al., 2025): MAE (Mean Absolute Error) and JSD (Jensen-Shannon Divergence). MAE quantifies the "quantity" error by comparing the average frequency of each substructure per molecule to the reference, while JSD assesses "distributional" similarity by measuring the statistical distance between the overall categorical proportions in the generated versus real datasets. Simply put, MAE ensures the correct amount of components per molecule, while JSD ensures the correct diversity across the entire chemical library. Furthermore, Fig. 9, 10, 11 display the distributions of bond lengths, bond angles, and torsion angles that occur frequently in CrossDock reference molecules.

EvoEGF-Mol demonstrates superior structural realism across all benchmarks (Tables 8,9,10). In atom type, ring size and functional groups tasks, our model achieves best results, which means EvoEGF-Mol closely mirrors the natural distribution of the CrossDock dataset. Notably, in functional group generation, the performance of EvoEGF-Mol indicates a high proficiency in recovering complex pharmacophores, which are often lost in noise-based generative processes. Regarding substructure distributions, EvoEGF-Mol is able to accurately reproduce the reference distributions, reflecting a deep capture of underlying geometric constraints.

*Table 8.* Frequency of the atoms on CrossDock.

| Atom Type | CrossDock | AR | Pocket2Mol | TargetDiff | DecompDiff | MolCRAFT | EvoEGF-Mol |
|---|---|---|---|---|---|---|---|
| C | 0.672 | 0.687 | 0.745 | 0.715 | 0.676 | 0.696 | 0.706 |
| N | 0.117 | 0.134 | 0.114 | 0.088 | 0.096 | 0.082 | 0.101 |
| O | 0.170 | 0.168 | 0.131 | 0.177 | 0.195 | 0.198 | 0.163 |
| F | 0.013 | 0.002 | 0.006 | 0.009 | 0.006 | 0.008 | 0.010 |
| P | 0.011 | 0.007 | 0.001 | 0.004 | 0.015 | 0.008 | 0.010 |
| S | 0.011 | 0.002 | 0.002 | 0.005 | 0.008 | 0.004 | 0.006 |
| Cl | 0.006 | 0.000 | 0.001 | 0.003 | 0.004 | 0.003 | 0.004 |
| MAE ($\downarrow$) | - | 0.849 | 0.842 | 0.411 | 0.398 | 0.273 | **0.176** |
| JSD ($\downarrow$) | - | 0.083 | 0.092 | 0.060 | 0.045 | 0.062 | **0.035** |

*Table 9.* Frequency of the rings on CrossDock.

| Ring Size | CrossDock | AR | Pocket2Mol | TargetDiff | DecompDiff | MolCRAFT | EvoEGF-Mol |
|---|---|---|---|---|---|---|---|
| 3 | 0.013 | 0.309 | 0.001 | 0.000 | 0.026 | 0.000 | 0.002 |
| 4 | 0.002 | 0.003 | 0.000 | 0.027 | 0.039 | 0.002 | 0.001 |
| 5 | 0.286 | 0.156 | 0.163 | 0.297 | 0.343 | 0.231 | 0.236 |
| 6 | 0.689 | 0.493 | 0.798 | 0.490 | 0.440 | 0.699 | 0.718 |
| 7 | 0.010 | 0.019 | 0.026 | 0.117 | 0.114 | 0.054 | 0.029 |
| 8 | 0.000 | 0.008 | 0.003 | 0.026 | 0.018 | 0.006 | 0.002 |
| MAE ($\downarrow$) | - | 0.256 | 0.146 | 0.161 | 0.198 | **0.074** | 0.076 |
| JSD ($\downarrow$) | - | 0.325 | 0.129 | 0.235 | 0.241 | 0.127 | **0.079** |

*Table 10.* Frequency of the top ten functional groups in Crossdock.

| Functional Group | CrossDock | AR | Pocket2Mol | TargetDiff | DecompDiff | MolCRAFT | EvoEGF-Mol |
|---|---|---|---|---|---|---|---|
| c1ccccc1 | 0.392 | 0.323 | 0.439 | 0.287 | 0.326 | 0.439 | 0.412 |
| NC=O | 0.147 | 0.179 | 0.097 | 0.141 | 0.156 | 0.166 | 0.121 |
| O=CO | 0.119 | 0.127 | 0.174 | 0.308 | 0.216 | 0.236 | 0.159 |
| c1ccncc1 | 0.045 | 0.077 | 0.090 | 0.057 | 0.052 | 0.025 | 0.056 |
| c1ncc2nc[nH]c2n1 | 0.034 | 0.013 | 0.001 | 0.000 | 0.008 | 0.000 | 0.003 |
| NS(=O)=O | 0.030 | 0.000 | 0.000 | 0.000 | 0.005 | 0.000 | 0.012 |
| O=P(O)(O)O | 0.022 | 0.071 | 0.010 | 0.016 | 0.038 | 0.033 | 0.030 |
| OCO | 0.019 | 0.083 | 0.025 | 0.084 | 0.045 | 0.030 | 0.013 |
| c1cncnc1 | 0.018 | 0.014 | 0.022 | 0.011 | 0.014 | 0.003 | 0.011 |
| c1cn[nH]c1 | 0.016 | 0.020 | 0.012 | 0.000 | 0.011 | 0.000 | 0.013 |
| MAE ($\downarrow$) | - | 0.047 | 0.031 | 0.044 | 0.032 | 0.026 | **0.020** |
| JSD ($\downarrow$) | - | 0.257 | 0.221 | 0.300 | 0.196 | 0.254 | **0.159** |

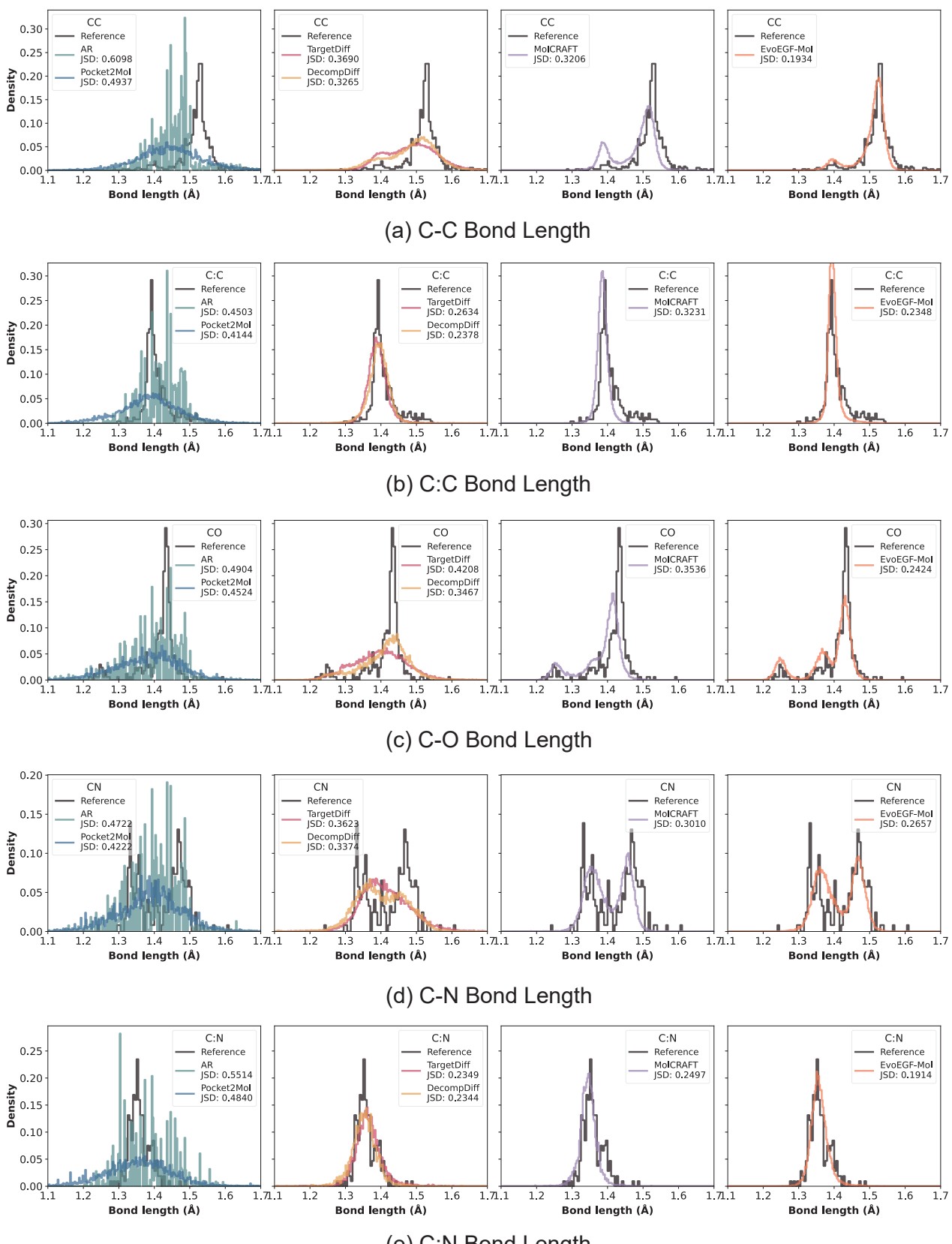

*Figure 9.* Comparison of the five most frequent bond length distributions between generated molecules and CrossDock references.

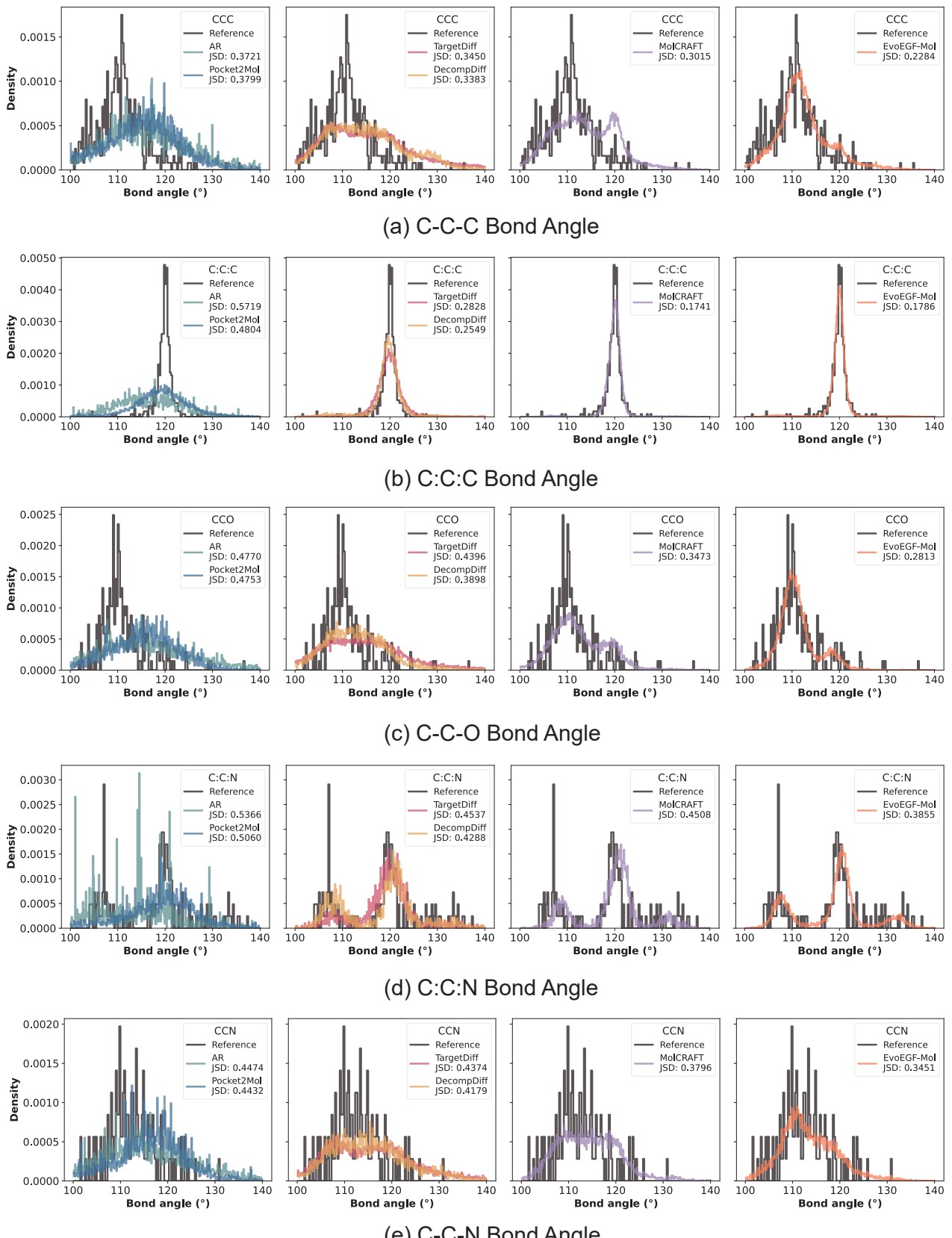

*Figure 10.* Comparison of the five most frequent bond angle distributions between generated molecules and CrossDock references.

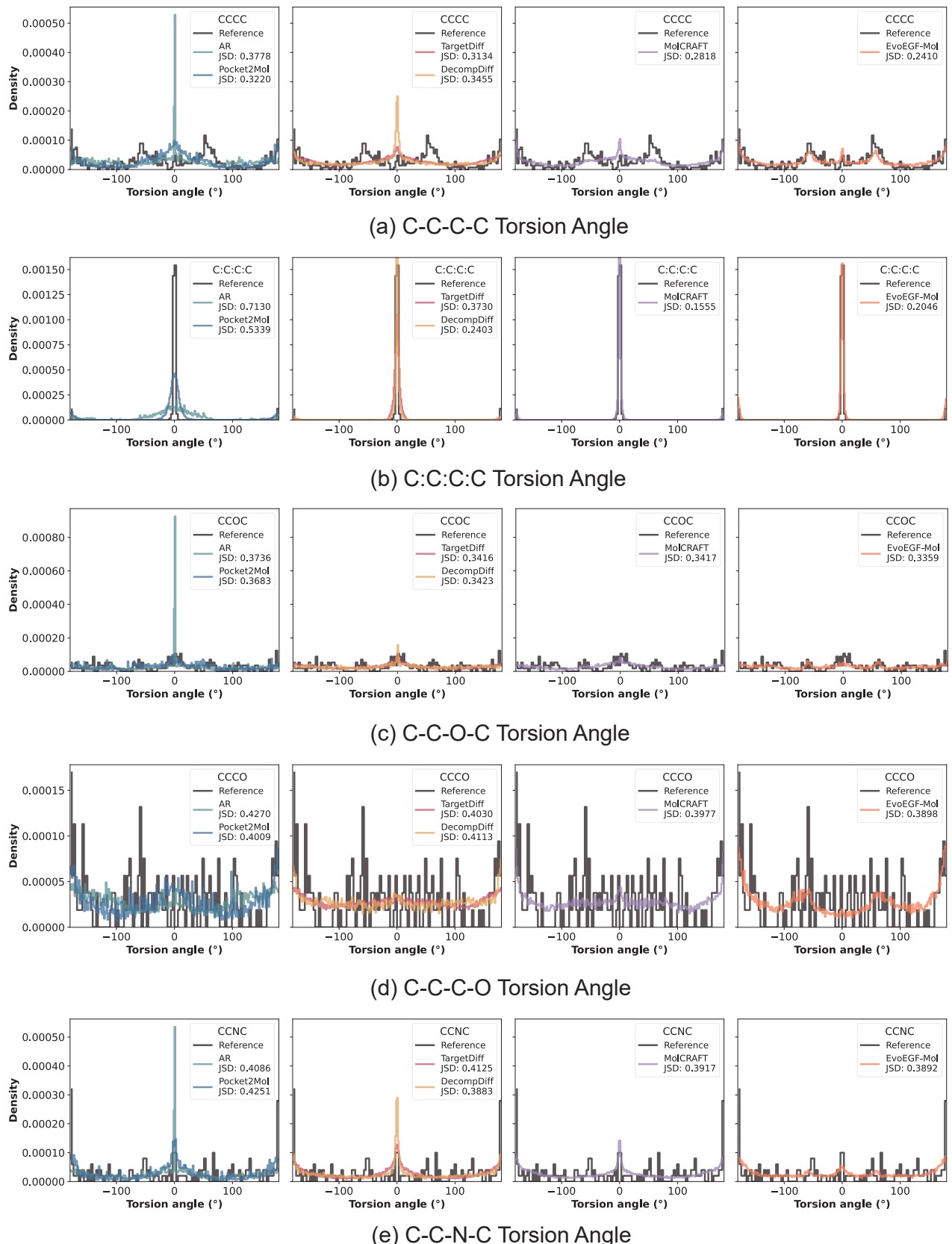

*Figure 11.* Comparison of the five most frequent torsion angle distributions between generated molecules and CrossDock references.

