# OpenReview forum: "EvoEGF-Mol: Evolving Exponential Geodesic Flow for Structure-based Drug Design"
_ICML.cc/2026/Conference — ICML 2026 regular_

### Official Review · Reviewer_gj6d · 2026-03-10

**Soundness:** 3
**Presentation:** 2
**Significance:** 3
**Originality:** 4
**Overall Recommendation:** 4
**Confidence:** 3

**Summary:**

The paper proposes to develop the flow matching on a statistical manifold with e-geodesic as the optimal transport path, to generate the structure and atom/bond type of molecules together, instead of using decoupled Cartesian and discrete flow matching. The downstream performance demonstrates the effectiveness of the proposed method.

**Compliance With Llm Reviewing Policy:**

Affirmed.

**Final Justification:**

My concerns have been adequately addressed.

**Key Questions For Authors:**

- Which baselines use the decoupled flow matching for structural and categorical modules? If not, I think the authors can add a ablation/Comparative experiments with decoupled flow matching to better support the key idea.

- Do the "intermediate states" generated at $t=0.5/ 0.6$ follow chemical rules (e.g., bond lengths and angles) more accurately than those from a decoupled flow model? Evidence of improved coupling between coordinates and atom types during the sampling process will be helpful.

**Limitations:**

yes

**Strengths And Weaknesses:**

Strengths:
- The idea that couples the different modules during molecular design is very interesting.
- The method is theoretically sound and also shows potential value in more general applications.

Weaknesses:
1. The writing can be improved. The motivation is “Conventional approaches construct probability paths separately in Euclidean and probabilistic spaces for continuous atomic coordinates and discrete chemical categories, leading to a mismatch with the underlying statistical manifolds.” However, the abstract and introduction didn’t directly and clearly state how the method coupled the coordinates and discrete chemical categories.

2. Since the model utilizes a more advanced geodesic flow, it should theoretically converge in fewer sampling steps $n$ than traditional linear flows. Providing a performance comparison across different values of $n$ can help demonstrate this efficiency.

3. Some crucial baselines are missing, like UniMOMO, FlexSBDD, and DynamicSBDD.  The improvement of the performance is marginal.

---

> ### Author Rebuttal · Authors · 2026-03-31
>
> Dear Reviewer,
>
> We sincerely thank you for your constructive feedback on our manuscript. We are encouraged that you found our approach of coupling different modules "very interesting", "theoretically sound", and of "potential value in more general applications". Below, we provide a response to your comments.
>
> # 1. Improvements to the Abstract and Introduction
> We agree that the coupling mechanism is central to our motivation and should be articulated more explicitly. We have revised abstract: we address this issue by modeling molecules as composite exponential-family distributions, where coordinates and categories are inherently coupled within a unified natural parameter space to evolve synchronously along Fisher-Rao geodesics.
> The revised introduction will be presented in the modified manuscript; it is omitted here due to space constraints.
>
> # 2. Sampling Efficiency
> Table 1 shows our model's robustness when reducing sampling steps ($n$): at $n=50$, it achieves 0.92 PB-Valid with low strain energy, outperforming baselines even at $n=20$.
>
> **Table 1: Model performance under different sampling steps**
>
> | Model | Vina Score (Med. $\downarrow$) | Vina Min (Med. $\downarrow$) | SE 25% ($\downarrow$) | SE 50% ($\downarrow$) | SE 75% ($\downarrow$) | $JS_{BL}$ ($\downarrow$) | $JS_{BA}$ ($\downarrow$) | QED ($\uparrow$) | SA ($\uparrow$) | PB-Valid (Avg. $\uparrow$) |
> | :--- | :--- | :--- | :--- | :--- | :--- | :--- | :--- | :--- | :--- | :--- |
> | EvoEGF-Mol-step20 | -6.42 | -6.76 | 18.53 | 53.71 | 129.57 | 0.22 | 0.31 | 0.53 | 0.67 | 0.86 |
> | EvoEGF-Mol-step50 | -6.76 | -7.04 | 11.7 | 32.91 | 71.58 | 0.22 | 0.29 | 0.53 | 0.73 | 0.92 |
> | EvoEGF-Mol-step100 | -6.89 | -7.12 | 8.94 | 25.96 | 56.65 | 0.22 | 0.28 | 0.53 | 0.75 | 0.93 |
>
> # 3. Baselines
>
> We omit DynamicFlow due to differing training data/re-training costs. Compared to FlexSBDD, EvoEGF-Mol sacrifices pocket flexibility (rigid treatment) but significantly improves geometric rationality. EvoEGF-Mol outperforms UniMoMo in small molecule generation metrics. We achieve SOTA-level performance comparable to MolPilot (see the response to Reviewer ZoUy). Despite the incremental affinity gains, the substantial leap in geometric validity and the ability to maintain high performance with 50% fewer inference steps represent a substantial practical advancement for real-world SBDD pipelines.
>
> **Table 2: Comparison of model performance with UniMoMo, FlexSBDD and MolPilot**
>
> | Model | Vina Score (Med. $\downarrow$) | Vina Min (Med. $\downarrow$) | Vina Dock (Med. $\downarrow$) | $JS_{BL}$ ($\downarrow$) | $JS_{BA}$ ($\downarrow$) | QED ($\uparrow$) | SA ($\uparrow$) | PB-Valid (Avg. $\uparrow$) |
> | :--- | :--- | :--- | :--- | :--- | :--- | :--- | :--- | :--- |
> | UniMoMo（all） | -5.72 | -6.08 | -7.25 | 0.32 | 0.38 | 0.55 | 0.7 | \ |
> | FlexSBDD | -7.2 | -8.73 | -9.54 | 0.3 | 0.31 | 0.58 | 0.69 | \ |
> | MolPilot | -7.03 | -7.27 | -7.92 | 0.23 | 0.29 | 0.56 | 0.74 | 0.96 |
> | EvoEGF-Mol | -6.89 | -7.12 | -7.88 | 0.22 | 0.28 | 0.53 | 0.75 | 0.93 |
>
> # 4. Comparisons on Decoupling
> Most baselines already utilize decoupled modeling for structural and categorical modules. Our experiments with TargetDiff and MolCRAFT effectively serve as controlled comparisons: since we use consistent architectures, the performance gains of EvoEGF-Mol (improved PB-Valid and strain) directly isolate the Fisher-Rao geodesic path as the key differentiator.
>
> # 5. Intermediate States
> Evaluating Validity (successful RDKit parsing into SMILES) and Completeness (parsing into a single, unified molecule) at early timesteps $t \in \{0.3, \dots, 0.6\}$ reveals that EvoEGF-Mol forms "molecular prototypes" significantly earlier than decoupled baselines (Table 3). Remarkably, the bond length and angle distributions of our intermediate structures at $t=0.6$ already exhibit higher fidelity than the final outputs ($t=1.0$) of TargetDiff and perform comparably to those of MolCRAFT (Table 4), confirming the superior chemical accuracy of our model's early-stage sampling.
>
> **Table 3: Comparison of intermediate states regarding validity and completeness**
>
> | Model | Validity($t=0.3$ $\uparrow$) | Validity($t=0.4$ $\uparrow$) | Validity($t=0.5$ $\uparrow$) | Validity($t=0.6$ $\uparrow$) | Completeness($t=0.3$ $\uparrow$) | Completeness($t=0.4$ $\uparrow$) | Completeness($t=0.5$ $\uparrow$) | Completeness($t=0.6$ $\uparrow$) |
> | :--- | :--- | :--- | :--- | :--- | :--- | :--- | :--- | :--- |
> | TargetDiff | 0.29 | 0.36 | 0.44 | 0.56 | 0.09 | 0.11 | 0.19 | 0.3 |
> | MolCRAFT | 0.78 | 0.9 | 0.96 | 0.99 | 0.37 | 0.6 | 0.86 | 0.92 |
> | EvoEGF-Mol | 0.96 | 0.97 | 0.98 | 0.99 | 0.39 | 0.75 | 0.94 | 0.97 |
>
> **Table 4: Comparison of intermediate states regarding bond length and bond angle distributions**
>
> | Model | $JS_{BL}$ ($\downarrow$) | $JS_{BA}$ ($\downarrow$) |
> | :--- | :--- | :--- |
> | TargetDiff (**$t=1$**) | 0.33 | 0.39 |
> | MolCRAFT (**$t=1$**) | 0.31 | 0.33 |
> | EvoEGF-Mol (**$t=0.6$**) | 0.31 | 0.34 |

---

> > ### Author Rebuttal · Reviewer_gj6d · 2026-04-02
> >
> > Thanks for the comprehensive response! I improved my score.

---

> > > ### Author Response · Authors · 2026-04-02
> > >
> > > We thank you for the positive assessment and for the insightful comments that helped refine our work. The discussed clarifications and additional details will be incorporated into the revised manuscript. We appreciate the time and effort dedicated to reviewing our submission.

---

### Official Review · Reviewer_e2sf · 2026-03-13

**Soundness:** 4
**Presentation:** 4
**Significance:** 3
**Originality:** 3
**Overall Recommendation:** 5
**Confidence:** 4

**Summary:**

The paper proposes EvoEGF-Mol, a generative framework for Structure-Based Drug Design (SBDD) that utilizes information-geometric flows on statistical manifolds. By modeling molecules as composite exponential-family distributions, the authors define generative flows along exponential geodesics (e-geodesics) to unify continuous coordinates and discrete chemical categories. To address the mathematical singularity and trajectory collapse caused by static Dirac targets, the model introduces a Dynamic Endpoint strategy that evolves from high-entropy states to data distributions over time.

**Compliance With Llm Reviewing Policy:**

Affirmed.

**Final Justification:**

All my concerns have been addressed.

**Key Questions For Authors:**

The formulation defines $\eta_{t}=(1-t)\eta_{0}+t\eta_{1}(t)$, where $\eta_{1}(t)$ is a time-evolving target.
Is it mathematically possible to achieve the same generative path by keeping the endpoint fixed at the data distribution ($\eta_{data}$) and simply reparameterizing the time schedule $t$?
Specifically, can we define a non-linear schedule $s(t)$ such that $\eta_{t} = (1-s(t))\eta_{0} + s(t)\eta_{data}$?
Please explain the unique advantages of explicitly evolving the target distribution parameters versus optimizing a non-linear interpolation schedule on a static target.

**Limitations:**

Yes

**Strengths And Weaknesses:**

Strengths
- Intuition and Clarity: The paper is clearly written, providing a strong intuitive motivation for bridging Euclidean and probabilistic spaces through the Fisher-Rao metric.
- Comprehensive Review: The authors provide a thorough overview of existing SBDD methods, effectively situating EvoEGF-Mol within the context of diffusion, Flow Matching, and BFN-based models.
- Robust Evaluation and Ablation: The model achieves impressive results, including a 93.4% PoseBusters passing rate on CrossDock. The ablation studies are extensive, covering the evolving mechanism, Dirichlet parameterization, and explicit bond diffusion.

Weaknesses
- Explanation of Metric Gains: While the model outperforms baselines in areas like Strain Energy and Vina Dock, more detailed reasoning is needed to explain how this specific information-geometric coupling mechanically results in such high geometric precision and interaction fidelity compared to conventional decoupled approaches.

---

> ### Author Rebuttal · Authors · 2026-03-31
>
> Dear Reviewer,
>
> We sincerely thank you for acknowledging the strong motivation and extensive ablation studies of EvoEGF-Mol. Your feedback on the mechanical reasoning and mathematical nuances of our approach has been invaluable in refining the manuscript. We address the specific comments below.
> # 1. Why EvoEGF-Mol Performs Well
> The superior performance of EvoEGF-Mol stems from three foundational design choices:
> ## 1.1 Auto-scaling of the Loss Function under the Fisher-Rao Metric
> Treating a molecule as a composite exponential family eliminates the need to manually balance incompatible loss weights (e.g., Euclidean MSE vs. Cross-Entropy).
> Under the Fisher-Rao metric, minimizing the local KL divergence between the ground-truth trajectory and the predicted trajectory naturally yields an additive loss function. The network optimizes an unweighted loss that is automatically scaled by the Fisher Information Matrices ($G^x$ and $G^v$): $$\mathcal{L} = \frac{1}{2} (\delta^x)^\top G^x(\eta^x_t) \delta^x + \frac{1}{2} (\delta^v)^\top G^v(\eta^v_t) \delta^v$$ Geometrically, optimizing this is equivalent to finding the geodesic path on a joint manifold, avoiding heuristic weight tuning.
> ## 1.2 The Alignment of Uncertainty
> We define specific certainty metrics for both variables:
> - Continuous (Coordinates $x$): Modeled as an isotropic Gaussian, where certainty is measured by precision $\Lambda_t$.
> - Discrete (Atom/Bond types $v$): Modeled as a Dirichlet distribution, where certainty is measured by the concentration parameter $\alpha_t$.
>
> By enforcing a shared linear evolution in the natural parameter space ($\eta_t$), both metrics follow identical trajectories:$$\eta_t = (1-t)\eta_0 + t\eta_1(t)$$This guarantees that despite differing in metric space and physical meaning, the "Information Progress Rate" for both variables remains perfectly synchronized.
>
> ## 1.3 Effective Learning Window via the Evolving Strategy
> Replacing static targets with a gradually evolving endpoint $\tilde{p}_1(t)$ prevents the "boundary singularity" where variance collapses instantly. This stretches the learning window across the entire time axis, ensuring the model effectively captures intricate cross-modal interactions.
>
> # 2. Does a non-linear schedule $s(t)$ exist?
> To determine if an evolving endpoint trajectory can be equivalently expressed as a reparameterized interpolation toward a fixed endpoint via a non-linear schedule $s(t)$, we analyze their mathematical equivalence.
> ## 2.1 Necessary Condition
> Evolving Endpoint: $\eta_t = (1 - t)\eta_0 + t\eta_1(t)$
> Reparameterized Path: $\eta_t = (1 - s(t))\eta_0 + s(t)\eta_{\text{data}}$
> Equating the two expressions requires that for any $t > 0$:$$\eta_1(t) = \eta_0 + \frac{s(t)}{t}(\eta_{\text{data}} - \eta_0)$$This means all $\eta_1(t)$ must lie on the same affine line defined by $\eta_0$ and $\eta_{\text{data}}$. Geometrically, the direction vectors $v(t) = \eta_1(t) - \eta_0$ must remain collinear across all $t$.
> ## 2.2 Gaussian Counterexample
> Using the 1D Gaussian exponential family, we construct an evolving endpoint where the target variance decays over time: $\sigma_1(t) = \lambda(1 - t)$.If we evaluate the ratio of the two components of $v(t)$ as the slope:
> $$R(t)=\frac{\eta^{(2)}_1(t)-\eta^{(2)}_0}{\eta^{(1)}_1(t)-\eta^{(1)}_0}=\frac{\frac{x^\*}{\lambda^2 (1-t)^2} - 0}{-\frac{1}{2\lambda^2 (1-t)^2} + \frac{1}{2}}=\frac{2x^\*}{-1 + \lambda^2(1-t)^2}.$$
>
> Since $R(t)$ is not constant, the direction of $v(t)$ changes. The trajectory curves and is not confined to a single affine subspace. Therefore, no scalar reparameterization $s(t)$ can equivalently represent this evolving path under the definition rule of such $\sigma_1(t)$.
>
> # 3. Advantages Over a Static Target with a Non-Linear Schedule
>
> We have discovered that probability paths defined by static endpoints combined with non-linear time scheduling form a subset of those defined by dynamic endpoints.
>
> Taking the expressions for $\eta_1(t)$ and $s(t)$ in Section 2.1 as an example, any time scheduling $s(t)$ can be mapped to a corresponding linear interpolation probability path defined by dynamic endpoints. However, certain probability paths defined by dynamic endpoints cannot be expressed through a single non-linear schedule $s(t)$, as demonstrated by the counterexample in Section 2.2. The reason is that dynamic endpoints allow for more flexible adjustments across different dimensions of the natural parameters, rather than forcing a single $s(t)$ to govern the scheduling of all natural parameters simultaneously.
>
> This suggests that designing probability paths using dynamic endpoints in the natural parameter space offers a broader expressive range (encompassing all possible paths using non-linear schedules). This provides a new perspective and expanded possibilities for future probability path design.

---

> > ### Author Rebuttal · Reviewer_e2sf · 2026-04-03
> >
> > Thanks for the reply.

---

> > > ### Author Response · Authors · 2026-04-04
> > >
> > > Thank you for your acknowledgement. We appreciate the time you have dedicated to reviewing our work and for the constructive dialogue during this rebuttal process.

---

### Official Review · Reviewer_ZoUy · 2026-03-13

**Soundness:** 3
**Presentation:** 3
**Significance:** 3
**Originality:** 2
**Overall Recommendation:** 3
**Confidence:** 3

**Summary:**

This paper proposes EvoEGF-Mol, a generative framework for structure-based drug design. The authors unify continuous variables and discrete variables of molecules into a composite exponential-family distribution and define generative flows along exponential geodesics on the Fisher-Rao statistical manifold. On both CrossDocked2020 and MolGenBench benchmarks, the model achieves superior molecular geometric plausibility and scaffold discovery metrics over the selected baselines.

**Compliance With Llm Reviewing Policy:**

Affirmed.

**Final Justification:**

I thank the authors for the response. EvoEGF-Mol uses MolPilot's network as its backbone but cuts the layers from 9 to 4 and adds gated attention. While the generative framework also differs from MolPilot, these changes only yield faster inference without improving key metrics such as PB-Valid (0.93 vs. 0.96). I will maintain my current score.

**Key Questions For Authors:**

Q1. How was the evolving endpoint hyperparameter $\lambda=0.2$ selected? Beyond the few values tested in ablation, was a broader search conducted? Does this value need to be adjusted for different protein targets?

**Limitations:**

yes

**Strengths And Weaknesses:**

**Strengths**

S1. The paper is well-organized with a natural flow between theoretical derivation and experimental analysis and the problem motivation is grounded in real-world constraints. Conventional methods model continuous coordinates and discrete atom types separately, causing a convergence mismatch: geometric coordinates converge while chemical identities remain undetermined. This modality mismatch is not a purely theoretical concern but a practical bottleneck affecting generated molecule quality. The ablation experiments confirm that the naive approach indeed fails in practice.


S2. The geometric quality of generated molecules is notably strong. PB-Valid reaches 93.4% and the median strain energy drops to 25.96, an order of magnitude lower than all baselines. The detailed 20-item PoseBusters breakdown shows strong performance across the board.

**Weaknesses**

W1. A relevant baseline is missing from the comparison. MolPilot (Qiu et al., 2025; "Piloting Structure-Based Drug Design via Modality-Specific Optimal Schedule," https://openreview.net/forum?id=b6VYI1Bvo2) reports 95.9% PB-Valid on the same CrossDocked2020 setup. A head-to-head comparison across the full set of metrics would substantially strengthen the empirical claims.


W2. Inference cost is not reported anywhere in the paper.


W3. The model generates only heavy atoms; hydrogens are added in post-processing. This is common practice in the field, but the paper does not acknowledge or discuss it as a limitation.

---

> ### Author Rebuttal · Authors · 2026-03-31
>
> Dear Reviewer,
>
> We sincerely thank you for your thoughtful and constructive feedback on our manuscript. We are encouraged that you found our work "well-organized" with a "notably strong" geometric quality, and that you recognized the practical value of addressing the modality mismatch in structure-based drug design. Below, we provide a point-by-point response to your comments.
>
> # 1. Comparison with MolPilot
>
> While MolPilot represents the current SOTA in SBDD, EvoEGF-Mol achieves highly competitive performance (Table 1) through a fundamentally different paradigm.
>
> MolPilot focuses on the temporal dimension, using dynamic programming to find an optimal off-diagonal path by decoupling noise schedules. Conversely, EvoEGF-Mol operates in the geometric dimension. By modeling molecules as exponential family distributions on a statistical manifold, we leverage the Fisher-Rao metric to define geodesic paths. This natively couples the information during generation through information geometry, rather than a decoupled search.
>
> **Table 1: Comparison of metrics between MolPilot and EvoEGF-Mol**
>
> | Model | PB-Valid (Avg. $\uparrow$) | Vina Score (Med. $\downarrow$) | Vina Min (Med.$\downarrow$) | Vina Dock (Med.$\downarrow$) | SE 25% ($\downarrow$) | SE 50% ($\downarrow$) | SE 75% ($\downarrow$) | $JS_{BL}$ ($\downarrow$) | $JS_{BA}$ ($\downarrow$) | QED ($\uparrow$) | SA ($\uparrow$) |
> | :--- | :--- | :--- | :--- | :--- | :--- | :--- | :--- | :--- | :--- | :--- | :--- |
> | MolPilot | 0.96 | -7.03 | -7.27 | -7.92 | 9.45 | 24.59 | 54.42 | 0.23 | 0.29 | 0.56 | 0.74 |
> | EvoEGF-Mol | 0.93 | -6.89 | -7.12 | -7.88 | 8.94 | 25.96 | 56.65 | 0.22 | 0.28 | 0.53 | 0.75 |
>
> Thinking about these generative tasks from a completely new perspective can bring:
> (1) Analytical Efficiency: MolPilot relies on post-hoc numerical searches (grid-discretization and DP). In contrast, EvoEGF-Mol offers an analytical solution via natural parameter interpolation. This bypasses the computational complexity and approximation errors inherent in MolPilot's optimization.
> (2) Conceptual Generalization: EvoEGF-Mol provides a principled framework for any mixed-modality generative task. It moves from empirical "path-finding" to rigorous "geodesic-flow" design on a unified statistical manifold.
>
>
> # 2. Inference cost
> As detailed in Appendix A.2 (line 600), our model uses 100 sampling steps, yielding inference costs comparable to MolCRAFT and MolPilot, and can generate 1.5 molecules per second on a 4090 GPU. Furthermore, reducing steps to 20 or 50 maintains robust performance (PB-Valid of 0.86 and 0.92, respectively, with persistently superior Strain Energy; see response to Reviewer f9RE). This ability to drastically reduce inference steps while maintaining stable, high-quality conformations demonstrates excellent computational efficiency.
>
> # 3. Adding hydrogens in post-processing
> We focus on heavy-atom generation to manage search space complexity, following standard SBDD protocols. Since hydrogen positions are reliably reconstructed via RDKit, this does not limit practical utility. We have clarified this implementation detail in the revised manuscript.
>
>
> # 4. The selection of hyperparameter $\lambda$
>
> We expanded our search to $\lambda \in \{0.5, 1.0\}$ (Table 2). Results confirm $\lambda=0.2$ balances training stability and geometric precision:
> - Small $\lambda$ ($\to 0$): Degenerates to a static Dirac distribution, causing trajectory collapse and numerical instability (Sec 3.2, line 197; App. B, line 635).
> - Large $\lambda$: Over-smooths the dynamic endpoint, forcing the model to mimic high-entropy distributions rather than precise geometries (evidenced by spiked Strain Energy).
>
> **Table 2: Model performance under different $\lambda$**
>
> | Model | Vina Score (Avg.$\downarrow$) | Vina Score (Med.$\downarrow$) | Vina Min (Avg.$\downarrow$) | Vina Min (Med.$\downarrow$) | SE 25% ($\downarrow$) | SE 50% ($\downarrow$) | SE 75% ($\downarrow$) | QED $\uparrow$ | SA $\uparrow$ |
> | :--- | :--- | :--- | :--- | :--- | :--- | :--- | :--- | :--- | :--- |
> | EvoEGF-Mol-$\lambda=0.5$ | -5.39 | -6.46 | -6.78 | -6.66 | 32.96 | 89.15 | 185.85 | 0.52 | 0.66 |
> | EvoEGF-Mol-$\lambda=1.0$ | -5.08 | -5.74 | -6.26 | -5.98 | 47.71 | 127.92 | 317.98 | 0.46 | 0.66 |
> | EvoEGF-Mol-$\lambda=0.2$ | -6.14 | -6.89 | -6.98 | -7.12 | 8.94 | 25.96 | 56.65 | 0.53 | 0.75 |
>
> $\lambda$ does not require per-target tuning. It regularizes the information-geometric path by mitigating pathological curvature on the statistical manifold rather than fitting target-specific features. A constant $\lambda$ generalizes effectively across diverse protein pockets.

---

> > ### Author Rebuttal · Reviewer_ZoUy · 2026-04-02
> >
> > Thanks for the response and the added experiments. According to Table 1, EvoEGF-Mol still falls behind MolPilot on PB-Valid (0.93 vs 0.96), which is the metric most directly tied to the paper's central argument about geometric plausibility. I would maintain my score.

---

> > > ### Author Response · Authors · 2026-04-03
> > >
> > > Dear Reviewer,
> > >
> > > Thank you for your continued engagement and your rigorous attention to the geometric plausibility of the generated molecules. You are absolutely correct that MolPilot achieves the strongest PB-Valid score of 0.96, and we respect this empirical result.
> > >
> > > While EvoEGF-Mol’s PB-Valid score (0.93) is slightly lower than MolPilot’s (0.96), it remains highly competitive within the SBDD field. We believe this marginal difference should be contextualized within the broader scope of our methodological contributions.
> > >
> > > Our goal with EvoEGF-Mol is not merely to chase an incremental SOTA on a specific benchmark, but to introduce a fundamentally different, mathematically principled paradigm to the community. To provide a more comprehensive view of our work, we would like to highlight the following key aspects:
> > >
> > > **1. Methodological Diversity and Theoretical Foundation**
> > > While MolPilot relies on a discrete, empirically optimized noise schedule (Dynamic Programming) to enforce validity, EvoEGF-Mol approaches the problem from a radically different angle: Information Geometry. We model the generative process via exponential geodesics on the Fisher-Rao statistical manifold. Achieving comparable, highly competitive results across the board *purely* through a continuous, analytical formulation—without extensive post-hoc schedule tuning—demonstrates the intrinsic power of this framework.
> > >
> > > Our performance gains over other diffusion-based baselines(like TargetDiff, MolCRAFT) using the identical network architecture and data directly isolate the Fisher-Rao geodesic as the key driver of improved geometric quality. We believe this theoretical advancement is valuable.
> > >
> > > The introduction of dynamic endpoints effectively mitigates the "boundary singularity" typical of static-target diffusion, stretching the critical learning window to better capture complex cross-modal interactions. Mathematically, this mechanism offers greater expressivity than simple non-linear time scheduling ($s(t)$); it allows trajectories to evolve along non-collinear paths on the statistical manifold (refer to the response to Reviewer e2sf for detailed analysis). In a sense, subsequent work can further design terminal evolution mechanisms or perform fine-tuning to achieve superior model performance.
> > >
> > > We believe that pushing the field of SBDD forward requires not only incremental metric gains but also the exploration of diverse, theoretically grounded methodologies that offer new heuristics.
> > >
> > > **2. Intrinsic Stability and Inference Flexibility**
> > > A key practical advantage of EvoEGF-Mol's analytical foundation is its robustness and flexibility during inference. Models reliant on DP-based optimal scheduling need to re-compute or re-tune their schedules if the number of inference steps changes. In contrast, the continuous nature of our exponential geodesic flow makes it inherently adaptable. As shown in our previous response to Reviewer gj6d, EvoEGF-Mol demonstrates remarkable stability, maintaining a strong PB-Valid of 0.92 even when inference steps are reduced by 50%. This structural robustness makes our method highly practical for real-world scenarios where inference efficiency matters.
> > >
> > > **3. Extensibility of the Framework**
> > > Because EvoEGF-Mol is built upon the fundamental principles of statistical manifolds rather than engineered heuristics, the framework is highly extensible. The analytical nature of our approach provides a solid mathematical foundation for future adaptations, such as integrating multi-objective optimization or applying it to different molecular modalities.
> > >
> > > We thank you for the feedback. We would like to clarify that the 0.93 PB-Valid is achieved through an analytical geodesic flow without the target-dependent numerical optimization required by MolPilot. We believe the mathematical elegance and sampling efficiency of EvoEGF-Mol, combined with its highly competitive geometric metrics, fulfill the central argument of our work. Our work aims to bridge this gap by offering a conceptually grounded approach that complements existing empirical efforts and provides new pathways for molecular generation.
> > >
> > > Thank you once again for your invaluable feedback, which has helped us to better articulate the core philosophy of our work.

---

### Official Review · Reviewer_f9RE · 2026-03-13

**Soundness:** 3
**Presentation:** 3
**Significance:** 3
**Originality:** 3
**Overall Recommendation:** 5
**Confidence:** 2

**Summary:**

This paper proposes EvoEGF-Mol, a structure-based molecular generation method that replaces standard parameter interpolation with an evolving exponential geodesic in the natural parameter space. The key idea is to use a dynamic endpoint whose variance/concentration is gradually sharpened over time, rather than aiming at a fixed delta-function target from the start. The method is evaluated on CrossDocked2020 and MolGenBench.

**Compliance With Llm Reviewing Policy:**

Affirmed.

**Final Justification:**

Thank you for your answer! I've changed the recommendation to "accept" from a "weak accept".

**Key Questions For Authors:**

1. Could the authors provide a clearer comparison against more recent strong SBDD generators, such as FLOWR and MolPilot, or explain why these comparisons were not included? Since the paper also cites methods like Pilot and DynamicFlow, it would be helpful to clarify which recent baselines are considered outside the intended comparison scope.

2. Am I right that the model is intended to produce physically valid ligand conformations at inference time without any explicit post-processing or correction step?

**Limitations:**

Yes

**Strengths And Weaknesses:**

**Strengths**:

The main idea is interesting and reasonably well motivated. The evolving-endpoint construction addresses a real issue of fixed sharp targets, and the ablation supports it: the naive fixed-endpoint variant collapses badly, with PB-Valid dropping to 64.1% and Vina Score becoming positive.

The strongest experimental evidence is on physical plausibility. On CrossDock, EvoEGF-Mol achieves the best PB-Valid (93.4%) and much lower strain-energy quartiles (8.94 / 25.96 / 56.65) than the baselines, while remaining competitive on docking metrics. Importantly, it is not uniformly best on affinity proxies: it is second to MolCRAFT on Vina Score and Vina Min, but best on Vina Dock. This makes the paper strongest as a contribution to more physically realistic generation, not necessarily to the best docking score overall.

The bond-generation ablation is also useful: removing explicit bond generation still gives a competitive Vina Score, but worsens PB-Valid and strain, supporting the claim that explicit bond modeling helps chemical integrity rather than only pocket fit.

**Weaknesses**:

My main concern is that the paper is much more convincing on geometry/physicality than on practical discovery utility. On MolGenBench scaffold metrics, the method is best among included baselines, but the paper also acknowledges that absolute hit rates remain low. At the molecule level, performance is still very limited: EvoEGF-Mol recovers 7 in-distribution hits, but 0 hits on proteins outside CrossDock, with hit fractions near 0.01%.

The comparison set is decent but not fully compelling. Baselines are taken from released samples in prior benchmark papers rather than rerun in a unified protocol, and the paper does not compare against some more recent methods it cites in related work. This weakens the empirical claim somewhat.

---

> ### Author Rebuttal · Authors · 2026-03-31
>
> Dear Reviewer,
>
> We sincerely thank you for your constructive feedback and for recognizing the physical plausibility and methodological motivation of EvoEGF-Mol. Your insights regarding practical discovery utility and baseline comparisons are invaluable for improving this work.
>
> We have carefully addressed each of your concerns as detailed below:
>
> # 1. Practical discovery utility
> We entirely agree that bridging the gap to practical discovery utility is an important goal of SBDD. We address the concern regarding low absolute hit rates below:
>
> (1). Dataset Limitations & Fair Benchmarking: Exact hit recovery is intrinsically limited by the CrossDock dataset, which contains computationally simulated complexes rather than experimentally validated bioactives. To rigorously evaluate our geometric framework, we restricted training to CrossDock to match baseline conditions. As shown in MolGenBench, all methods trained exclusively on CrossDock exhibit near-zero exact hit recovery. Introducing external 2D bioactive datasets would have confounded this architectural comparison.
>
> (2). Geometric Plausibility as a Prerequisite: A model's ability to discover hits in SBDD is fundamentally bottlenecked by its ability to navigate the complex, non-Euclidean manifold of molecular structures. By resolving the premature variance collapse inherent in traditional exponential geodesics, EvoEGF-Mol provides a reliable geometric engine. This guarantees stringent physical constraints (evidenced by our 93.4% PoseBusters pass rate), establishing a physically grounded "higher floor" for generative quality, even if absolute hit rates remain a community-wide challenge.
>
> (3). Future Work: With this geometric foundation secured, scaling the model for practical discovery utility is our immediate next step. We will train EvoEGF-Mol on robust, bioactive datasets to deploy it within a real-world drug design pipeline.
>
> # 2. Evaluation Protocol
> Regarding the baseline evaluation, we ensured a unified protocol by using the same code and scoring methodologies as those used for the original benchmarks. Since the metrics for all baselines and our proposed method were calculated identically, the comparison is consistent and the empirical results remain reliable despite not re-generating the samples.
>
> # 3. Baseline Model Selection
>
> While MolPilot represents the current SOTA in SBDD, EvoEGF-Mol achieves competitive performance through a distinct conceptual approach. A specific analysis is provided in our response to Reviewer ZoUy.
> MolPilot decouples noise schedules temporally, using dynamic programming to find optimal paths on the loss surface.
> EvoEGF-Mol operates geometrically, utilizing the Fisher-Rao metric to construct probability paths on the statistical manifold, naturally coupling structural and chemical information.
>
> Due to FLOWR’s custom data splits, a full re-evaluation is computationally prohibitive within the rebuttal period. However, direct comparison using their reported metrics on CrossDock (Table 1) shows that EvoEGF-Mol performs on par with or slightly better than FLOWR. Notably, as discussed with Reviewer gj6d, our model demonstrates superior stability in low-NFE (sampling step) regimes.
>
> When bonds are not explicitly utilized (identical training data and architecture as TargetDiff and MolCRAFT), EvoEGF-Mol consistently outperforms both across most indicators(line 283,284,1108). This highlights the inherent mathematical advantages of the EvoEGF framework.
>
> **Table 1: Comparison of model performance with FLOWR**
>
> | Model | PB-Valid (Avg. $\uparrow$) | Vina Score (Avg. $\downarrow$) | Vina Min (Avg. $\downarrow$) |
> | :--- | :--- | :--- | :--- |
> | FLOWR | 0.92 | -6.29 | -6.48 |
> | EvoEGF-Mol | 0.93 | -6.14 | -6.98 |
>
> Furthermore, we have included additional comparisons with UniMoMo and FlexSBDD in our response to Reviewer gj6d.
>
> # 4. Clarification on the Inference Stage
> You are correct in understanding. The model is designed to generate physically valid ligand conformations directly without requiring any explicit post-processing steps such as bond assignment or force-field optimization. The only minor step involved is the addition of hydrogen atoms, as the model primarily generates heavy-atom framework. This is a common convention in the field. Importantly, since our model explicitly generates bond information, the hydrogen placement is more straightforward and accurate.

---

> > ### Author Rebuttal · Reviewer_f9RE · 2026-04-03
> >
> > Thank you for your answer! I've changed the recommendation to "accept" from a "weak accept"

---

> > > ### Author Response · Authors · 2026-04-03
> > >
> > > We are grateful for your positive reassessment and the valuable suggestions provided. The final manuscript will incorporate the improvements discussed during the rebuttal. Thank you for your time and for helping us refine this work.

---

### Decision · Program_Chairs · 2026-04-30

**Decision:**

Accept (regular)

**Comment:**

Reviewers were broadly positive about both the conceptual contribution and the empirical evidence. In particular, they found the evolving-endpoint geodesic flow formulation well motivated, and they considered the physical plausibility results and ablations especially convincing. The paper also appears to be clearly written, and at least one reviewer explicitly improved their recommendation after rebuttal. One reviewer remained more skeptical about how much novelty lies beyond prior backbones, but this concern did not outweigh the broader positive consensus reflected in the final scores.